# Controllable Video Synthesis via Variational Inference

## Abstract

Many video workflows benefit from a mixture of user controls with varying granularity, from exact 4D object trajectories and camera paths to coarse text prompts, while existing video generative models are typically trained for fixed input formats. We develop a video synthesis method that addresses this need and generates samples with high controllability for specified elements while maintaining diversity for under-specified ones. We cast the task as variational inference to approximate a composed distribution, leveraging multiple video generation backbones to account for all task constraints collectively. To address the optimization challenge, we break down the problem into step-wise KL divergence minimization over an annealed sequence of distributions, and further propose a context-conditioned factorization technique that reduces modes in the solution space to circumvent local optima. Experiments suggest that our method produces samples with improved controllability, diversity, and 3D consistency compared to prior works.

## 1 Introduction

Recent developments in video generative models (Wang et al., 2025; Brooks et al., 2024; Ho et al., 2022; Blattmann et al., 2023; Yang et al., 2024; Chen et al., 2025b) have enabled great capabilities in content generation (Jiang et al., 2025), robot learning (Yang et al., 2023; Du et al., 2023b; Bharadhwaj et al., 2024), world modeling (Bruce et al., 2024), and other applications. While exhibiting impressive visual quality, two limitations exist: most models only provide fixed-form user interfaces, such as text and first-frame prompts, to describe desired content; in addition, scene inconsistency or drifting (Zhang & Agrawala, 2025a) exacerbates with longer-duration generation. Our goal is to develop methods to faithfully follow a spectrum of user controls from easy-to-specify but coarse ones to harder-to-specify but precise ones, including texts, background contexts in the form of image or video prompts, camera trajectories, and simulated assets and trajectories (Fig. 1). Doing so will provide versatile user interfaces and improve output scene consistency and physical fidelity, addressing both of the aforementioned limitations.

Existing works extending the capabilities of video generative models can be categorized into two: approaches with scalable data curation strategies and model finetuning (Tu et al., 2025; Jiang et al., 2025), and approaches applied during inference time to steer generation outputs with external scalar rewards via search heuristics (Liu et al., 2025). This work falls into the latter and can benefit from better end-to-end models from the first category. With the popularity of diffusion and flow models (Lipman et al., 2022; Liu et al., 2022; Sohl-Dickstein et al., 2015), several theoretical frameworks (Skreta et al., 2025; He et al., 2025; Wu et al., 2023b) propose to steer and compose such models using importance weight correction to ensure unbiasedness, but remain to be empirically validated on video data. On the other hand, prior compositional methods (Du et al., 2023a; Zhang et al., 2025) typically use Langevin MCMC (Parisi, 1981; Welling & Teh, 2011) for sampling and are exact only in the limit of infinite simulation steps.

Inspired by these approaches, we cast our task into a sampling problem from a target distribution composed of individual ones (Hinton, 1999; Genest & Zidek, 1986; Du & Kaelbling, 2024), each corresponding to one constraint or desired property, and construct an annealed sequence of distributions (Neal, 2001; Kirkpatrick et al., 1983) as matching targets to ease sampling. To maintain sample diversity without suffering from weight degeneracy (Skreta et al., 2025) or inefficiency from MCMC mixing, we take a variational approach and optimize with Stein Variational Gradient De-

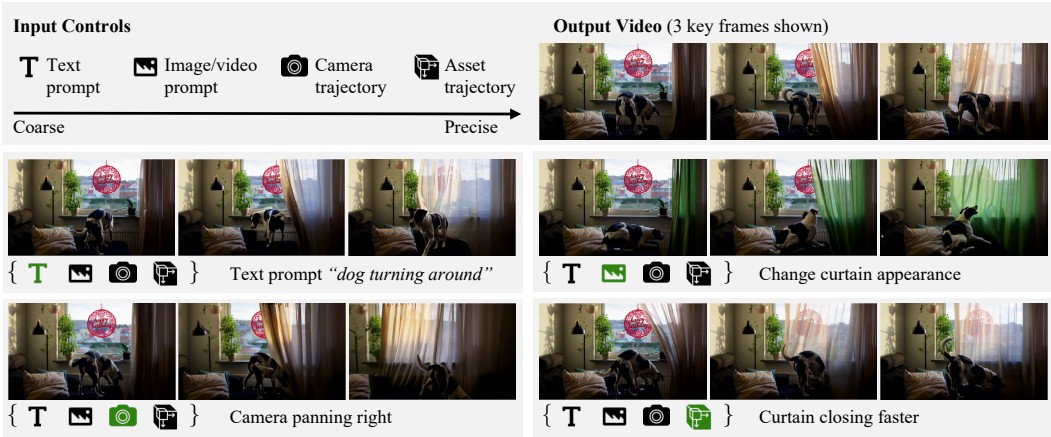

Figure 1: **Task Overview.** This work develops a controllable video synthesis framework, where different forms of user inputs (top left) are supported and can be flexibly mixed for each generation pass. Top-right shows an output sample. Users can further change any prompt element to achieve different levels of control over outputs (bottom).

scent (SVGD) (Liu & Wang, 2016), minimizing the KL divergence between an empirical distribution of particles and each intermediate target distribution before achieving the final target distribution.

During optimization, particles move towards high-density regions via deterministic transport, which is susceptible to local optima. To alleviate this, we propose a 3D-aware conditioning technique that augments the target with auxiliary conditionals, effectively simplifying the distribution and, in addition, improving 3D consistency in generated outputs, as validated in experiments in standard fixed-length video generation and in longer-duration generation settings.

Our contributions are summarized below:

1. We propose a video synthesis framework accepting mixed forms of controls for each generation, ranging from coarse text prompts to precise camera trajectories and 4D assets.
2. We develop a variational inference method with simulated annealing, together with a conditioning strategy tailored for video data for efficient inference.
3. On controllable video synthesis tasks, the framework improves visual fidelity, output diversity, and scene consistency over prior works.

## 2 RELATED WORKS

**Controllable Video Generation.** Most prior works on conditional video generation focus on a fixed set of modalities of user conditions (Brooks et al., 2024; Wang et al., 2025; Chen et al., 2025b; He et al., 2024; Geng et al., 2025). Recent works tackle the controllable video synthesis task by training multi-condition video generation models with large-scale data curation (Jiang et al., 2025; Tu et al., 2025). Our work builds on top of these methods to support a more flexible control interface.

**Compositional Generative Modeling.** Recent work explores compositional generative modeling (Du & Kaelbling, 2024; Du et al., 2020; Garipov et al., 2023; Huang et al., 2022a; Mahajan et al., 2024; Bradley et al., 2025; Thornton et al., 2025; Gaudi et al., 2025; Zhang et al., 2025; Du et al., 2023a; Jung et al., 2024; Wu et al., 2023c; Akan & Yemez, 2025b;a), where multiple models are combined to jointly generate data samples. These works mostly use MCMC-based (Robert et al., 1999) sampling, while our work uses a variation inference strategy to trade off unbiasedness with better sampling efficiency, which is critical for high-dimensional data like videos, and develops techniques specific to the application of interest.

**Variational Inference for Diffusion Models.** Variational inference methods approximate a target distribution by typically minimizing the divergence measure between an optimizable, tractable distribution and the target (Blei et al., 2017; Liu & Wang, 2016; Kingma & Welling, 2013). Within

---

**Algorithm 1** Video Synthesis with Annealed Variational Inference

---

1: **Inputs:** Text prompt $\mathcal{Y}$, image $\mathcal{I}$, camera trajectory $\mathcal{C}$; pre-trained models $\{p^{(i)}(\cdot)\}_{i=1}^N$; annealing length $T$ and schedule $\{t_T, \cdots, t_0\}$; particle count $L$; SVGD step size $\eta$; kernel $k$.
2: **Init particles:** $\{x^{(l)}\}_{l=1}^L, x^{(l)} \sim \mathcal{N}(0, I)$                  $\triangleright$ Sample from $q_T$
3: **Preprocessing:** Compute masks $\mathcal{M}^{(i)}$ and context conditionals $\{z_t^{\text{context}}\}_{t=T}^0$
4: **for** $t = T - 1$ to $0$ **do**                        $\triangleright$ Anneal $p_{t_{t+1}} \Rightarrow p_t$
5:      **for** $l = 1$ to $L$ **do**                 $\triangleright$ Parallel over particles
6:          Compute score $s_t^*(x^{(l)})$ with Eq. (6) and then velocity $v_t^*(x^{(l)})$ using Eq. (S2)
7:          $x^{(l)} \leftarrow x^{(l)} + v_t^*(x^{(l)})/T$          $\triangleright$ Initialization with Eq. (S6)
8:          $\phi^*(x^{(l)}) = \frac{1}{L} \sum_{l'=1}^L \left[ k(x^{(l')}, x^{(l)}) \nabla_{x^{(l')}} \log p_t^*(x^{(l')}) + \nabla_{x^{(l')}} k(x^{(l')}, x^{(l)}) \right]$
9:          $x^{(l)} \leftarrow x^{(l)} + \eta \phi^*(x^{(l)})$                  $\triangleright$ SVGD step
10:         $x^{(l)} \leftarrow (1 - \mathcal{M}_{\text{context}}) \odot x^{(l)} + \mathcal{M}_{\text{context}} \odot z_t^{\text{context}}$ $\triangleright$ Correction with context conditionals
11:         (Optional) Update $x^{(l)}$ with gradient $\nabla_{x^{(l)}} \| \mathcal{M}_{\text{context}} \odot \hat{x}_0(x^{(l)}) - z_0^{\text{context}} \|_2^2$, where $\hat{x}_0(x^{(l)})$ is the clean prediction from Eq. (S3)
12:      **end for**
13:      Refine masks $\mathcal{M}^{(i)}$ (Section 3.2)
14: **end for**
15: **Output:** Video samples $\{x^{(l)}\}_{l=1}^L$.

---

the context of diffusion models, it has been adapted to solve inverse problems (Song et al., 2023; Mardani et al., 2023)) and to interpret training objectives (Huang et al., 2021; Kingma & Gao, 2023; Kingma et al., 2021), while in this work, we focus on practical video synthesis applications.

## 3 METHOD

Given input images and texts, 3D assets and their pose sequences, and camera trajectories, our goal is to generate a video that adheres to the input prompts, contains the assets under the specified poses, and has the input image as the background for the initial frame. Examples are shown in Fig. 1. We will first introduce the proposed algorithm in Section 3.1 and then its implementation in Section 3.2.

### 3.1 ALGORITHM

**Objective.** Let $x \in \mathbb{R}^d$ be video data. Given $N$ pre-trained models $\{p^{(i)}\}_{i=1}^N$ and their inputs $y = \left(y^{(i)}\right)_{i=1}^N$, we aim to approximate an intractable product distribution $p^*$ by finding a distribution $q$ with the following KL objective (Blei et al., 2017; Kingma & Welling, 2013):

$$\mathcal{J}(q) = \text{KL}(q\|p^*), \quad \text{where } p^*(x \mid y) :\propto \prod_i p^{(i)}(x \mid y^{(i)}). \tag{1}$$

Direct optimization of the above objective is intractable. Inspired by simulated annealing methods (Song & Ermon, 2019; Neal, 2001), we construct a sequence of annealed distributions $\{p_t^*(x \mid y)\}_{t=T}^0$, and solve a sequence of intermediate minimization problems under per-step objectives

$$\mathcal{J}(q_t) = \text{KL}(q_t\|p_t^*), \quad \text{where } p_t^*(x \mid y) :\propto \prod_i p_t^{(i)}(x \mid y^{(i)}), \tag{2}$$

and $p_t^{(i)}$ are marginal distributions of the backbone flow models (Section B.1).

**Optimization.** We set the starting distribution $q_T := p_T = \mathcal{N}(0, I)$. For each annealing step $t < T$, we employ Stein Variational Gradient Descent (SVGD) (Liu & Wang, 2016), a non-parametric method that represents $q_t$ with a set of particles $\{x^{(l)}\}_{l=1}^L$ and applies an analogy of variational gradient descent to $q_t$ to solve the optimization problem in Eq. (2).

We initialize $q_t$ by transporting particles from $q_{t+1}$ one Euler step along predicted velocity $v_t^*$ (Section B.2). Given a kernel $k : \mathbb{R}^d \times \mathbb{R}^d \to \mathbb{R}$, Liu & Wang (2016) considers a function $\mathbf{f} : \mathbb{R}^d \to \mathbb{R}^d$ within a unit ball in the associated reproducing kernel Hilbert space (RKHS), and identifies the

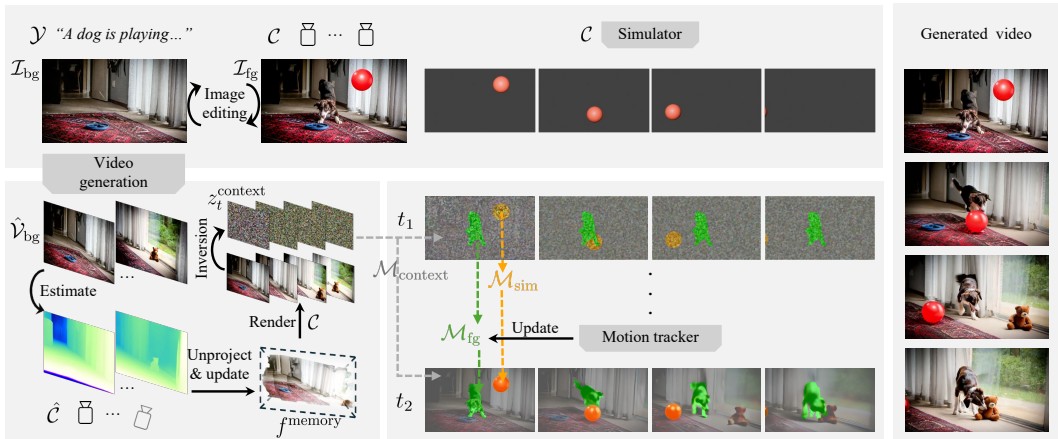

Figure 2: **Method Overview. Top-left:** Input specification, including text prompt $\mathcal{Y}$, camera trajectory $\mathcal{C}$, input image pair $\{\mathcal{I}_{\text{bg}}, \mathcal{I}_{\text{fg}}\}$, and 3D asset trajectory. **Bottom-left:** We then compute context conditionals $z_t^{\text{context}}$ that provide background priors for maintaining scene consistency. **Bottom-center:** Foreground masks $\mathcal{M}_{\text{fg}}$ and simulation masks $\mathcal{M}_{\text{sim}}$ define the regions handled by their respective models, while $\mathcal{M}_{\text{context}}$ specifies the region for context conditionals. Showing one particle for simplicity. **Right:** Example frames from the generated video.

following functional gradient evaluation in analytic form: $\nabla_{\mathbf{f}} \text{KL}((I + \mathbf{f})_\sharp q_t \| p_t^*)|_{\mathbf{f}=0} = -\phi^*(x)$, where $\sharp$ is the pushforward operator and $\phi^*(\cdot) := \mathbb{E}_{x \sim q_t(x)} \left[ k(x, \cdot) \nabla_x \log p_t^*(x)^T + \nabla_x k(x, \cdot) \right]$. This means that out of all local perturbations $(I + \mathbf{f})|_{\mathbf{f}=0}(x)$ being considered, $\mathbf{f}^* = \eta \phi^*$ with a small step size $\eta \in \mathbb{R}$ results in the maximum decrease in KL divergence. This result can be directly used to transport current particles (lines 9 from Algorithm 1). Examining $\phi^*$, the first term pushes particles toward high-density regions of the target, and the second prevents particles from collapsing.

**Factoring with Context Conditionals.** The deterministic update described above is susceptible to local optima. We therefore propose to sample from a factorized distribution:

$$p_t^*(x \mid y) \propto \prod_i \mathbb{E}_{z \sim p(z|y^{(i)})} \left[ p_t^{(i)}(x \mid y^{(i)}, z) \right] \quad \text{(law of total probability)} \tag{3}$$

$$\approx \prod_i p^{(i)}(x \mid y^{(i)}, z^*) \quad \text{where } z^* = \arg\max_z p(z \mid y) \tag{4}$$

$$\propto \prod_i p_t^{(i)}(x \mid y^{(i)}) p(z^* \mid x, y^{(i)}). \quad \text{(Bayes' rule)} \tag{5}$$

Here, $z$ is a variable introduced to reduce modes from each intermediate target distribution $p_t^*$. Integration over $z$ in Eq. (3) is intractable, and we approximate it using MAP estimation over complete inputs $y$ (Murphy, 2012). For example, $z$ corresponds to a plausible background video given all input descriptions. Additional terms $p(z^* \mid x, y^{(i)})$ in Eq. (5) can be interpreted as additional models that regularize the product distribution. Running the optimization in Eq. (2) requires computing scores for Eq. (5), which is available as Eq. (6) under a particular choice of $z^*$ to be introduced in Section 3.2. Each $p^{(i)}(x)$ corresponds to a target distribution specified by pretrained backbones.

## 3.2 FRAMEWORK INSTANTIATION

The rest of this section instantiates the framework for controllable video synthesis tasks. Denote a video sample as $x \in \mathbb{R}^d$ with $d = H \times W \times N \times 3$; $H, W$ is the spatial resolution and $N$ is the number of RGB frames. User input consists of an image $\mathcal{I}$ describing the scene context, a text prompt $\mathcal{Y}$, a camera trajectory parameterized as a sequence of $N$ camera matrices $\mathcal{C} \in \mathbb{R}^{N \times 4 \times 4}$, and a video of a simulated object $\mathcal{V}_{\text{sim}}$ rendered with $\mathcal{C}$.

We assume access to a text-image-to-video model $f^{\text{image}}(x, t) = \nabla_x \log p_t(x \mid \mathcal{I}, \mathcal{Y})$, and video generation models conditioned on low-level visual maps, $f^\psi(x, t) = \nabla_x \log p_t(x \mid \psi(\mathcal{V}_{\text{sim}}), \mathcal{Y})$, where $\psi \in \{\psi^{\text{depth}}, \psi^{\text{flow}}, \psi^{\text{traj}}\}$ denotes a depth, optical flow, or object trajectory extractor. The

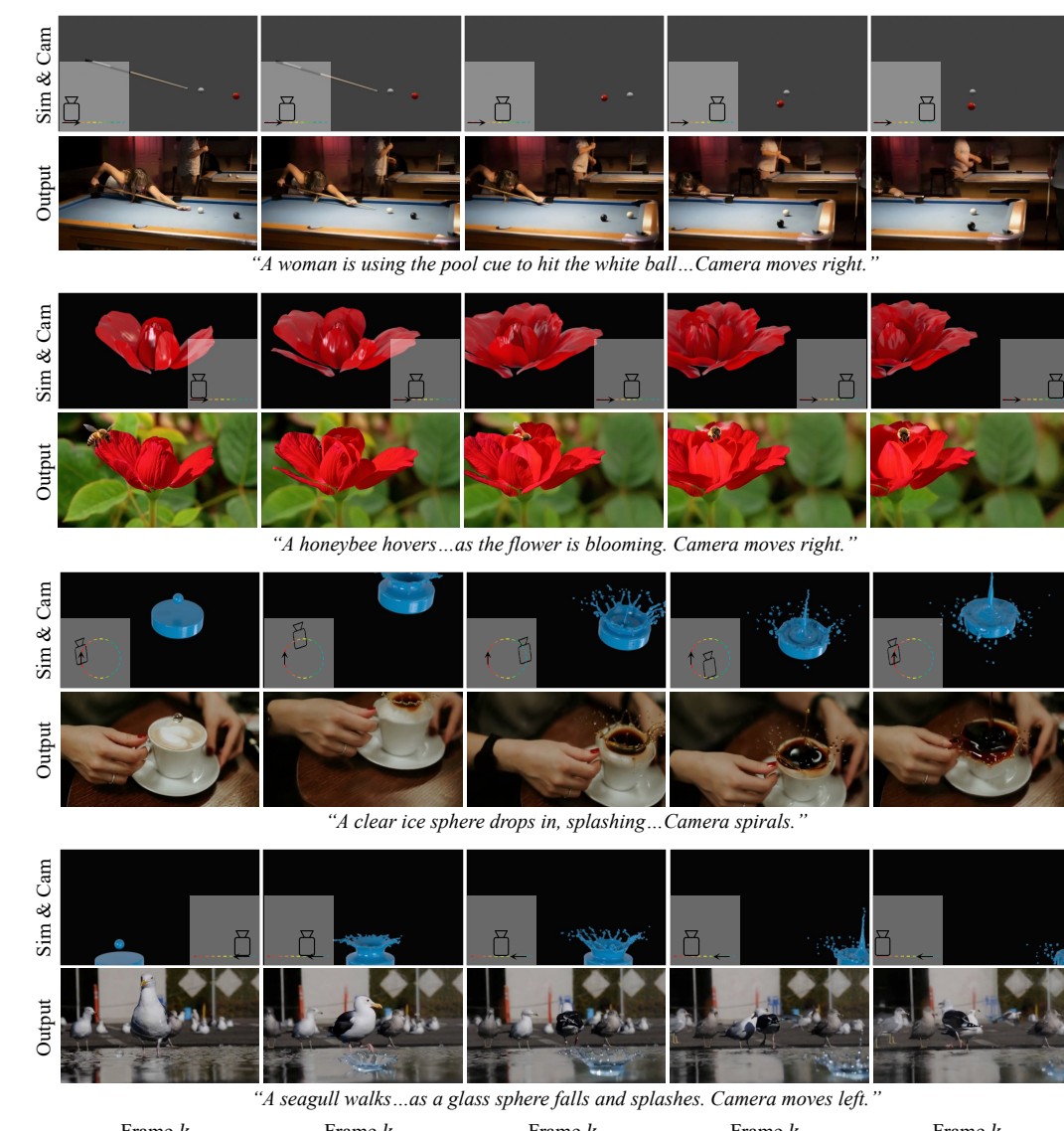

Figure 3: **Qualitative Results.** For each test case, input object simulation and camera trajectory are visualized, with text prompts shown at the bottom. Our method generates videos aligned with object and camera trajectories while exhibiting natural and coherent content for unconstrained regions.

intermediate target distribution from Eq. (5) amounts to a (weighted) product distribution with score

$$s_t^*(x) := \nabla_x \log p_t^*(x \mid y) = \mathcal{M}_{\mathrm{fg}} \odot f^{\mathrm{image}}(x) + \mathcal{M}_{\mathrm{sim}} \odot f^\psi(x) + \mathcal{M}_{\mathrm{context}} \odot (-\lambda(x - z_t^{\mathrm{context}})), \quad (6)$$

where spatially varying masks $\mathcal{M}^{(i)} \in [0, 1]^{H \times W \times N \times 1}$ and scalar $\lambda \in \mathbb{R}$ reflect the relative importance of the corresponding backbone models. The choice $z^* := z_t^{\mathrm{context}}$ together with computation of other related quantities will be expanded below. Derivations are deferred to Section B.3.

**Preprocessing.** We first construct a foreground-background image pair $(\mathcal{I}_{\mathrm{fg}}, \mathcal{I}_{\mathrm{bg}})$ through image editing (Section C.2). The input image $\mathcal{I}$ may be the foreground or background image itself, and the foreground content can be either underspecified—allowing intricate object motions and diverse synthesis—or precisely constrained via 4D control signals such as depth or optical flow extracted from simulation, corresponding to different variants of user interfaces (Fig. 1).

**Computing Context Conditionals.** To implement Eq. (5), $z^*$ is computed from a reference video $\mathcal{V}_{\mathrm{bg}}$ following the assigned camera trajectory $\mathcal{C}$ for a scene determined by $\mathcal{I}_{\mathrm{bg}}$. Specifically, we use a camera-to-video model $f^{\mathrm{camera}}$ to generate $\hat{\mathcal{V}}_{\mathrm{bg}}$ conditioned on $\mathcal{C}$ and $\mathcal{I}_{\mathrm{bg}}$. Due to the inaccuracy of

| Methods | Controllability | | | Video Quality | | | | Semantic Alignment | |
|---|---|---|---|---|---|---|---|---|---|
| | LPIPS (↓) | Camera (↑) | GPT-4o (↑) | Subject (↑) | Photo (↑) | 3D Consist (↑) | GPT-4o (↑) | ViCLIP (↑) | GPT-4o (↑) |
| Image2V | 0.086±0.069 | 0.521±0.346 | 0.695±0.258 | 0.633±0.071 | 0.788±0.198 | 0.726±0.201 | 0.709±0.251 | 0.203±0.045 | 0.614±0.268 |
| Depth2V | 0.078±0.069 | 0.550±0.360 | 0.673±0.260 | 0.606±0.048 | 0.868±0.102 | 0.816±0.103 | 0.705±0.249 | 0.219±0.026 | 0.591±0.250 |
| Flow2V | 0.075±0.051 | 0.439±0.400 | 0.686±0.249 | 0.613±0.062 | 0.786±0.187 | 0.748±0.268 | 0.709±0.244 | 0.214±0.040 | 0.609±0.232 |
| Cam2V | 0.089±0.075 | 0.668±0.304 | 0.655±0.274 | 0.617±0.051 | 0.764±0.170 | 0.774±0.136 | 0.691±0.254 | 0.202±0.049 | 0.577±0.284 |
| GEN3C | 0.101±0.069 | **0.853±0.134** | 0.677±0.236 | 0.575±0.070 | **0.886±0.049** | **0.904±0.059** | 0.714±0.238 | 0.208±0.058 | 0.573±0.215 |
| PoE-I&D | 0.078±0.055 | 0.430±0.405 | 0.650±0.221 | 0.600±0.056 | 0.749±0.140 | 0.649±0.368 | 0.682±0.237 | 0.211±0.041 | 0.555±0.196 |
| PoE-C&D | 0.072±0.042 | 0.636±0.284 | 0.702±0.234 | 0.575±0.070 | 0.727±0.106 | 0.798±0.024 | 0.709±0.102 | 0.215±0.034 | 0.632±0.217 |
| Ours | **0.068±0.030** | 0.791±0.246 | **0.773±0.248** | **0.634±0.077** | 0.791±0.087 | 0.896±0.279 | **0.755±0.249** | **0.221±0.043** | **0.682±0.241** |

Table 1: **Controllable Video Generation.** View synthesis method GEN3C excels in 3D consistency but lags in other dimensions, while video generative models show better controllability and visual quality but weaker geometric consistency. Our method achieves better or comparable performance compared to these two classes of methods. We further conduct a scaled-up evaluation in Table S1.

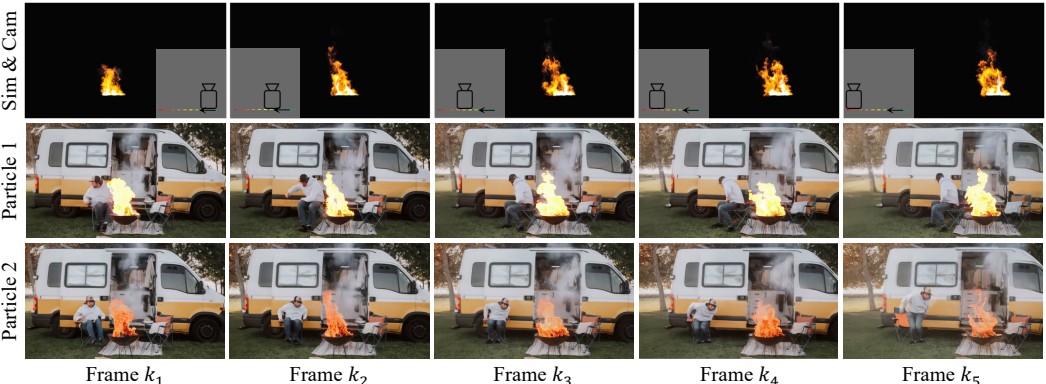

*"The flames suddenly flare with sparks and the man reacts in shock. Camera moves left."*

Figure 4: **Output Diversity.** A set of particles is obtained during the proposed optimization procedure for population-level sampling. The above contains visualizations for two particles from the same optimization. They both follow the input conditions (fire trajectory and text prompts), while presenting diversity in under-specified regions, e.g., human.

$f^{\text{camera}}$, there exists a discrepancy between the desired and actual camera trajectory in the generated video. We therefore query a dynamic novel view synthesis model GEN3C (Ren et al., 2025) $f^{\text{memory}}$ with $\hat{\mathcal{V}}_{\text{bg}}$ and and $\mathcal{C}$ to re-render the final reference video $\mathcal{V}_{\text{bg}}$, and run inversion (Wang et al., 2024a) to compute $(z_t^{\text{context}})_{t=T}^0$ which will be used in Eq. (6). See Section C.3 for details.

**Weighting with Adaptive Masks.** The foreground mask $\mathcal{M}_{\text{fg}} \in \mathbb{R}^{H \times W \times N}$ is computed by duplicating the semantic segmentation mask for unconstrained foreground objects from input foreground image $\mathcal{I}_{\text{fg}}$ across $N$ frames, $\mathcal{M}_{\text{sim}} \in \mathbb{R}^{H \times W \times N}$ is available from the simulator, and finally, $\mathcal{M}_{\text{context}} := (1 - \mathcal{M}_{\text{fg}}) \odot (1 - \mathcal{M}_{\text{sim}})$. During denoising process, $\mathcal{M}_{\text{fg}}$ is updated to be the predicted mask from a dynamic motion tracker Huang et al. (2025) for the currently generated video, once the unconstrained objects exhibit sufficiently clear dynamics for tracker to estimate. To enforce fidelity of the background, we minimize the reconstruction loss on background pixels (line 11, Algorithm 1).

# 4 EXPERIMENTS

We empirically evaluate the proposed method in this section with standard, fixed-length (Section 4.1) and with extended length (Section 4.2), followed by ablations (Section 4.3).

## 4.1 CONTROLLABLE VIDEO SYNTHESIS

**Evaluation Details.** We evaluate our method against baselines including image-to-video (Image2V), depth-to-video (Depth2V), flow-to-video (Flow2V), and camera-to-video (Cam2V), all built on Wan2.1 or Wan2.2 (14B). We further compare with a novel view synthesis model GEN3C (Ren et al., 2025) and a compositional generation method PoE (Zhang et al., 2025) with two

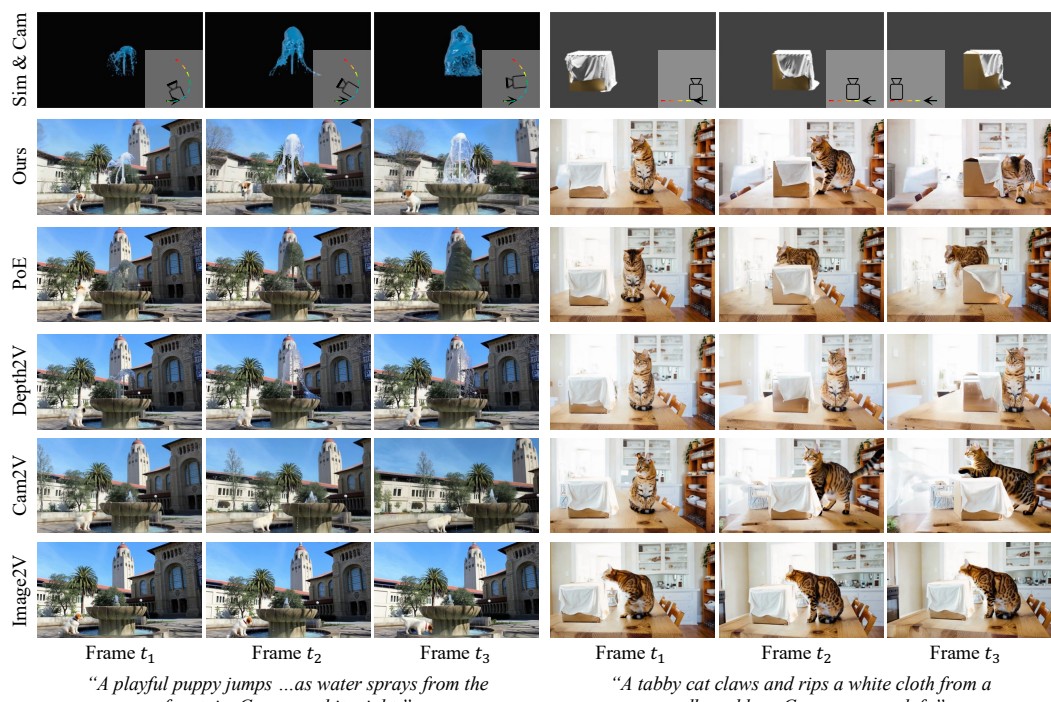

Figure 5: **Baseline Comparisons.** Failure modes of baselines include fixed cameras under out-of-distribution trajectories (e.g., "orbit right" on the left) and unnatural object drifting (right), while ours achieves better controllability by incorporating constraints from multiple model components.

| Methods | Controllability | | | Video Quality | | | | Semantic Alignment | |
|---|---|---|---|---|---|---|---|---|---|
| | LPIPS (↓) | Camera (↑) | GPT-4o (↑) | Subject (↑) | Photo (↑) | 3D Consist (↑) | GPT-4o (↑) | ViCLIP (↑) | GPT-4o (↑) |
| GEN3C | 0.120±0.046 | **0.839**±0.182 | 0.825±0.083 | 0.542±0.037 | 0.882±0.048 | 0.861±0.087 | 0.825±0.083 | 0.232±0.025 | 0.700±0.071 |
| FramePack | 0.136±0.046 | 0.535±0.387 | 0.750±0.109 | 0.576±0.070 | **0.948**±0.030 | 0.859±0.059 | **0.899**±0.071 | 0.226±0.042 | 0.750±0.166 |
| SkyReels | 0.138±0.047 | 0.549±0.351 | 0.650±0.218 | 0.555±0.071 | 0.602±0.169 | 0.494±0.204 | 0.750±0.150 | 0.213±0.041 | 0.575±0.259 |
| Wan2.2 | 0.129±0.044 | 0.504±0.312 | 0.602±0.013 | **0.614**±0.055 | 0.797±0.162 | 0.709±0.188 | 0.700±0.010 | 0.237±0.033 | 0.489±0.008 |
| w/o MSE | 0.102±0.041 | 0.713±0.135 | 0.837±0.022 | 0.553±0.036 | 0.611±0.004 | 0.674±0.185 | 0.763±0.054 | 0.244±0.019 | 0.787±0.114 |
| Ours | **0.099**±0.039 | 0.815±0.224 | **0.925**±0.043 | 0.606±0.013 | 0.897±0.162 | **0.862**±0.029 | 0.875±0.083 | **0.248**±0.039 | **0.849**±0.112 |

Table 2: **Controllable Video Synthesis with Extended Length.** Generated videos are of 8s, 11s, or 14s duration. Our method can be directly applied to this task, achieving better or comparable performance compared to prior video synthesis methods.

variants: PoE-I&D (using Image2V and Depth2V) and PoE-C&D (using Cam2V and Depth2V). We construct the evaluation dataset with 10 simulations, 8 camera trajectories, and 13 scenes, yielding 50 test samples in total. Each test case is evaluated with 2 samples per method.

Evaluation metrics cover controllability, video quality, and semantic alignment. We calculate the LPIPS (Zhang et al., 2018) distance between outputs and simulator RGB renderings on regions of simulated objects, and adopt 4 metrics from WorldScore (Duan et al., 2025): Camera Controllability (Camera), Subjective Quality (Subject), 3D Consistency (3D Consist), and Photometric Consistency (Photo), which is to assess the appearance consistency across frames. Text prompt alignment is measured using ViCLIP (Wang et al., 2023). We query GPT-4o for automatic evaluation with more details in Section C.5. All metrics are normalized to range [0, 1]. We also evaluate our method on standard benchmark FullBench (Ju et al., 2025) with details in Section D.

**Results.** As shown in Table 1, our method achieves 3D consistency and camera controllability comparable to the 3D-aware GEN3C. It surpasses all baselines in visual quality and follows both semantic and physical simulation instructions more faithfully. Compared to PoE approaches, our method yields lower LPIPS and better semantic alignment. This indicates better preservation of constrained object dynamics without reducing diversity in unconstrained regions. We further conduct a scaled-up evaluation with 200 samples in Table S1, which shows the same overall trends.

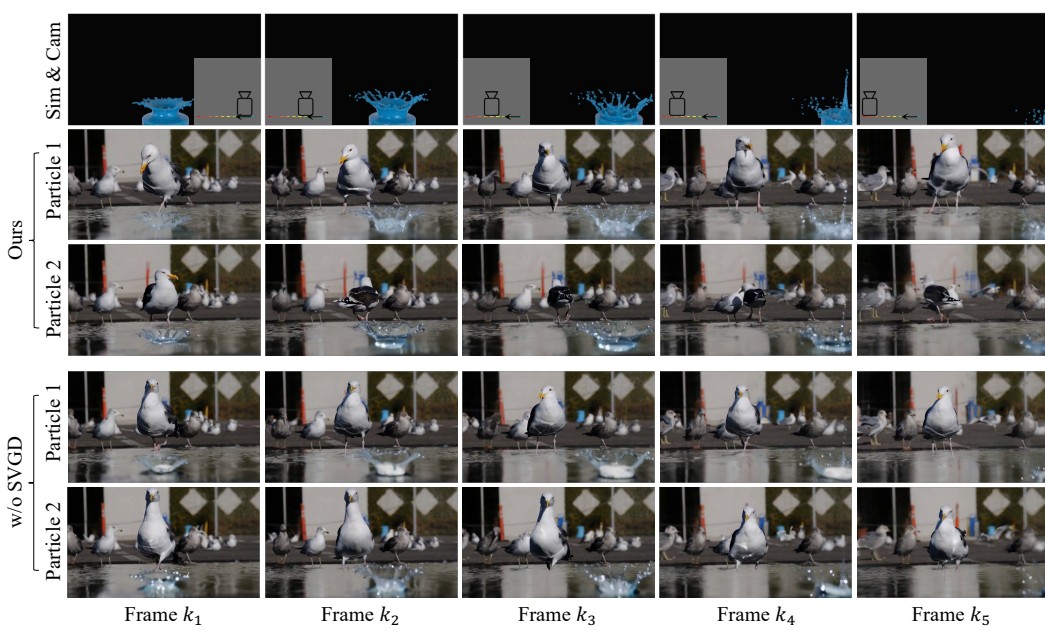

*"A seagull walks…as a glass sphere falls and splashes. Camera moves left."*

Figure 6: **Ablation on SVGD.** SVGD promotes greater variation in unconstrained regions (e.g., the seagull), enabling richer and more varied generations without sacrificing quality.

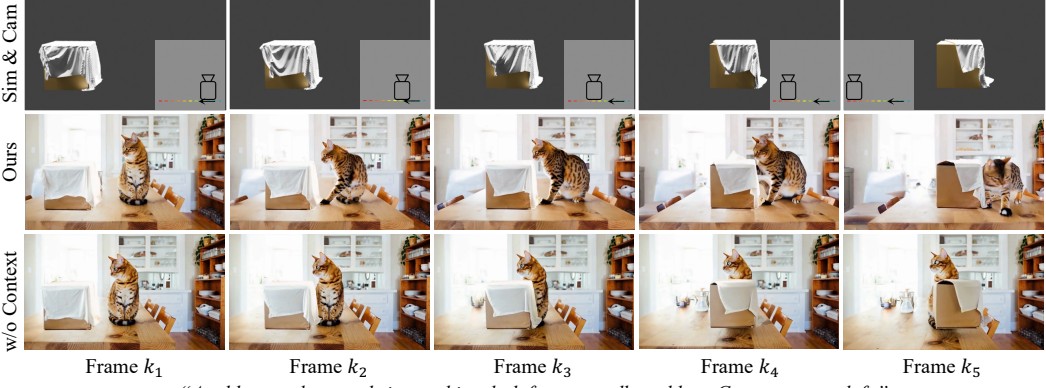

*"A tabby cat claws and rips a white cloth from a cardboard box. Camera moves left."*

Figure 7: **Ablation on Context Conditionals.** Without context conditionals during sampling, the model is prone to local optima, such as cardboard box sliding unnaturally across a static table.

Qualitative comparisons in Fig. 5 further show that prior video models, including Cam2V model, fail to reliably follow camera trajectories. This problem is most evident in out-of-distribution cases such as "orbit right" on the left of Fig. 5. In contrast, our model successfully handles more challenging scenarios, including the "camera spirals" example shown in the third panel of Fig. 3.

In addition, Fig. 4 shows that our method produces diverse yet faithful samples. As discussed in Ablation Section 4.3, this diversity arises from SVGD updates, which encourage particles to explore different modes of the distribution while remaining consistent with the constraints.

## 4.2 VIDEO SYNTHESIS WITH EXTENDED LENGTH

While the backbone video models used in our implementation produce a fixed number of frames by default, our framework naturally extends to longer-duration generation while mitigating common issues of content drifting and scene inconsistency.

**Implementation Details.** We divide long videos into overlapping temporal segments, each containing 81 frames. We maintain a 3D memory cache that stores the context conditionals across

| Method | SSIM↓ | LPIPS↑ | PSNR↓ |
|---|---|---|---|
| w/o SVGD | 0.754±0.051 | 0.279±0.041 | 20.947±2.246 |
| Ours | **0.740**±0.043 | **0.298**±0.031 | **20.160**±0.982 |

Table 3: **Output Diversity** measured via similarity among particles. Less similarity is better as it corresponds to higher diversity. Results show that SVGD contributes positively to sample diversity.

| Methods | Controllability | | | Video Quality | | | | Semantic Alignment | |
|---|---|---|---|---|---|---|---|---|---|
| | LPIPS (↓) | Camera (↑) | GPT-4o (↑) | Subject (↑) | Photo (↑) | 3D Consist (↑) | GPT-4o (↑) | ViCLIP (↑) | GPT-4o (↑) |
| w/o Context | 0.082±0.059 | 0.446±0.397 | 0.673±0.257 | 0.575±0.070 | **0.812**±0.148 | 0.718±0.244 | 0.724±0.235 | 0.211±0.051 | 0.582±0.234 |
| w/o SVGD | 0.071±0.051 | **0.801**±0.177 | 0.771±0.029 | **0.636**±0.056 | 0.765±0.101 | **0.905**±0.264 | **0.775**±0.029 | 0.220±0.048 | **0.682**±0.109 |
| Ours | **0.068**±0.030 | 0.791±0.246 | **0.773**±0.248 | 0.634±0.077 | 0.791±0.087 | 0.896±0.279 | 0.755±0.249 | **0.221**±0.043 | **0.682**±0.241 |

Table 4: **Ablations.** Removing context conditionals (w/o Context) degrades 3D consistency and camera controllability. Disabling SVGD maintains quality but reduces diversity (Table 3).

segments. After each segment is generated, the last $K = 33$ frames are reused as context conditionals for the overlapping portion of the next segment. We further apply inversion and reconstruction loss to enforce smooth transitions and temporal coherence. Details are provided in Section C.4.

**Evaluation.** We compare our method with the backbone model Wan2.2 and novel view synthesis model GEN3C (Ren et al., 2025), as well as two representative autoregressive video generative models, SkyReels-V2 (Chen et al., 2025b) and FramePack (Zhang & Agrawala, 2025b), which are natively suitable for longer-context generation. Wan2.2 (Wang et al., 2025) is implemented by reusing the last frame of a generated clip as the first frame of the next clip and concatenating the outputs. Experiments are conducted on 4 scenes selected from the dataset in Section 4.1, yielding 20 test cases with varying video lengths of 8s, 11s, and 14s, corresponding to 2, 3, and 4 temporal segments, respectively, to reflect different difficulty levels. Each test case is evaluated with 2 samples per method. We report the same metrics as in Section 4.1.

**Results.** Results in Table 2 show that our method achieves the best semantic alignment while maintaining high controllability. It also reaches comparable camera controllability and 3D consistency compared to GEN3C, which is trained specifically for 3D-aware tasks. The ablation further demonstrates that reconstruction loss significantly improves both geometric (3D Consist) and appearance consistency (Photo) in long video generation. Qualitative samples are shown in Fig. S2.

### 4.3 ABLATION STUDIES

**SVGD.** Multiple particles ($L = 2, 4, 8$) are jointly updated via SVGD. In the ablated variant, SVGD optimization among particles is disabled, and only a single particle is propagated forward. Results in Table 3 and Fig. 6 suggest that SVGD updates improve sample diversity while maintaining comparable visual quality (Table 4). Without SVGD, outputs are susceptible to local optima, whereas with SVGD, the particle updates encourage diverse yet high-fidelity generations. We provide more details and discussions in Section F.

**Context conditionals.** We ablate the effect of context conditionals by comparing the full method with a variant skipping line 10 in Algorithm 1. Results in Table 4 suggest that context conditionals play critical role in preserving geometry and enforcing camera controllability. The observation is reflected qualitatively in Fig. 7, where the ablated variant results in drifts and background distortions.

**Memory and Runtime.** We compare memory and run-time against baselines. Details in Section G.

## 5 CONCLUSION

We introduced a controllable video synthesis framework that supports a spectrum of user inputs, including text and image prompts, camera trajectories, and asset trajectories. We cast the task in a variational inference framework in an annealed optimization scheme for efficient sampling, and propose a 3D-aware context conditioning technique to address the optimization challenge and improve output scene consistency. Experiments demonstrate that our method achieves favorable performance measured with controllability, diversity, and scene consistency, extending the capabilities of video generation models to be more powerful creation tools.

**Reproducibility Statement.** We provide full implementation details of our method, as well as evaluation protocols in Section 4.1, Section 4.2, and Section C. We include experimental results in Section 4 and supplementary materials. To facilitate reproducibility and future research, we will publicly release our full codebase and evaluation dataset upon acceptance.

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

## A  OVERVIEW

This appendix provides additional details and results to supplement the main paper. It contains algorithm details (Section B), implementation details (Section C), more experimental results (Section E and Section F), computation requirement (Section G) and limitation discussion (Section H). Video results are included in the local webpage `index.html`.

## B  ALGORITHM DETAILS

### B.1  FLOW MODELS

For flow-based models (input constraints are dropped for simplicity) with Gaussian paths:

$$p_\tau(\cdot) = \int p_{\tau|1}(\cdot \mid x_1) p_1(x_1) \, dx_1, \quad p_{\tau|1}(\cdot \mid x_1) := \mathcal{N}\left(\cdot \mid \alpha_\tau x_1, \sigma_\tau^2 I\right), \quad x_1 \sim p_{\text{data}}(x_1), \quad \text{(S1)}$$

where $(\alpha_\tau, \sigma_\tau)_{\tau \in (0,1)}$ is a continuous noise schedule. In the main paper, $t = T, T-1, T-2, \cdots, 0$ is implemented as its discretization with corresponding to $\tau = 0, 1/(T+1), 2/(T+1), \cdots, 1$. Typically, $\alpha_0 = 0, \sigma_0 = 1$ and $\alpha_1 = 1, \sigma_1 = 0$, and therefore $p_0 = \mathcal{N}(0, I)$.

To compute line 6 in Algorithm 1, velocity can be computed from

$$v_\tau(x) = \frac{\dot{\alpha}_\tau}{\alpha_\tau} x - \frac{\dot{\sigma}_\tau \sigma_\tau \alpha_\tau - \dot{\alpha}_\tau \sigma_\tau^2}{\alpha_\tau} s_\tau(x), \quad \text{(S2)}$$

To compute line 11, the clean prediction for $x$ from timestep $\tau$:

$$\hat{x}_0(x) = \frac{\sigma_\tau}{\dot{\alpha}_\tau \sigma_\tau - \dot{\sigma}_\tau \alpha_\tau} v_\tau(x) - \frac{\dot{\sigma}_\tau}{\dot{\alpha}_\tau \sigma_\tau - \dot{\sigma}_\tau \alpha_\tau} x. \quad \text{(S3)}$$

### B.2  INITIALIZATION

Algorithm 1 maintains a set of $L$ particles $\{x^{(l)}\}_{l=1}^L$.

$$q_t := \sum_l \delta_{x^{(l)}}. \quad \text{(S4)}$$

For $t < T$, $q_t$ is initialized from

$$q_t^{\text{init}} = \mathcal{K}_\sharp q_{t+1}, \quad (\mathcal{K}_\sharp q_{t+1})(\cdot) \stackrel{\text{def}}{=} \int \mathcal{K}_{t \leftarrow t+1}(\cdot \mid x) q_{t+1}(dx), \quad \text{(S5)}$$

where $\mathcal{K}_{t \leftarrow t+1}(\cdot \mid x) = \delta_{x + v_t^*(x)/T}(\cdot)$ transports samples from the noise level $t+1$ to $t$ following predicted velocity $v_t^*(x)$

Plugging in Eq. (S4) to Eq. (S5):

$$q_t^{\text{init}} = \sum_l \delta_{x^{(l)} + v_t^*(x^{(l)})}. \quad \text{(S6)}$$

This corresponds to the initialization step in line 8, Algorithm 1.

### B.3  SCORES OF ANNEALED PRODUCT DISTRIBUTION

Taking the log for both sides for Eq. (5):

$$s_t^*(x) = \nabla_x \log p_t^*(x \mid y) = \sum_i (\nabla_x \log p_t^{(i)}(x \mid y^{(i)}) + \nabla_x \log p(z^* \mid x, y^{(i)})). \quad \text{(S7)}$$

The above score function computation is used in Eq. (2) and requires computing $\nabla_x \log p(z^* \mid x, y^{(i)}))$, which we approximate as a direct delta distribution centered at $x$, corresponding to a hard constraint such that $x$ is assigned with nonzero density under $\log p_t^*(x \mid y)$ only at $x = z^*$. To prevent $s_t^*(x)$ from infinite values, we first compute $s_t^*(x)$ with $\lambda = 0$ (line 6 from Algorithm 1), followed by a projection step (line 10).

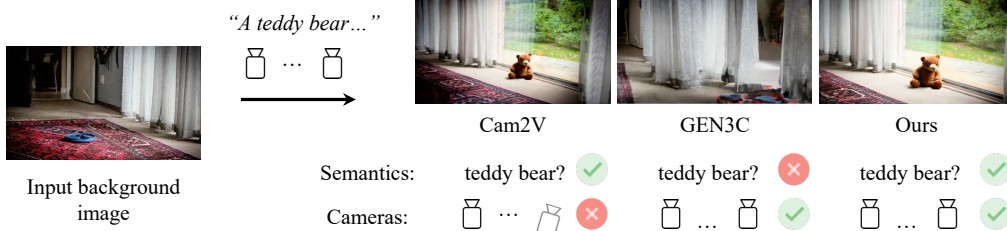

Figure S1: **Computing Context Conditionals.** Camera-conditioned video generative models (Cam2V) better preserve semantic content (e.g., inserting the teddy bear) but often fail to follow the exact camera trajectory. In contrast, 3D-aware models (GEN3C) ensure accurate camera control but may miss semantic instructions and introduce artifacts in unobserved regions. We combine both by first generating semantically rich background using Cam2V, then re-rendering it with GEN3C to produce $\mathcal{V}_{bg}$ as context conditionals.

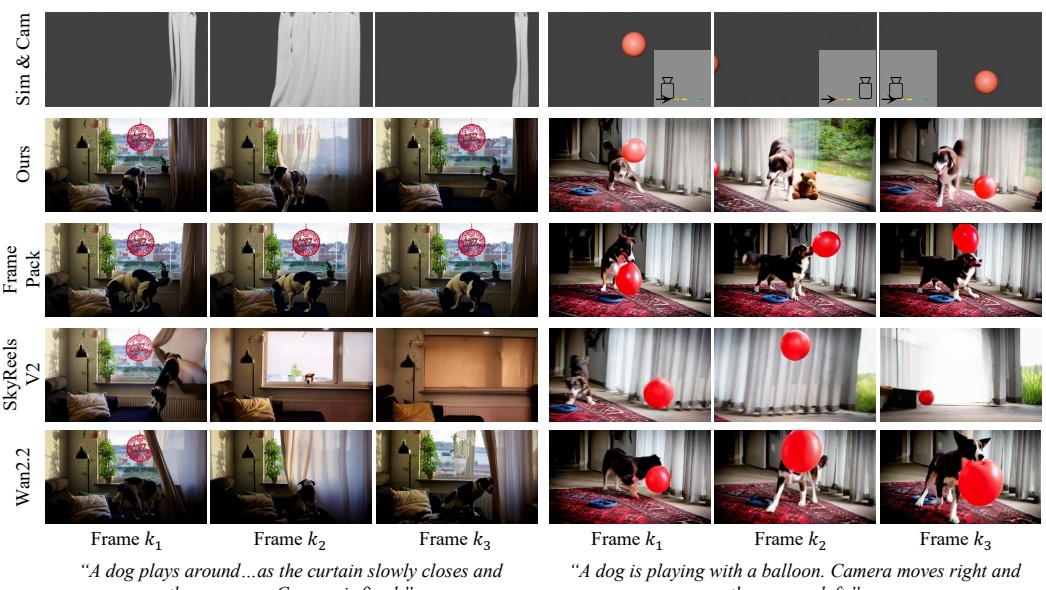

Figure S2: **Baseline Comparisons on Controllable Video Synthesis with Extended Length.** Our method can effectively preserve contextual details from earlier segments. For example, it retains the scenery outside the window and preserves the decorative window patterns after the curtain closes and reopens (left panel). In addition, the method maintains scene consistency when the camera returns to previously visited scenes (right panel).

### B.4 GENERATING REFERENCE VIDEO

We observed that models trained on text-to-video synthesis and novel view synthesis tasks have complementary capabilities: the former tend to preserve semantic information in generated outputs while suffering from content forgetting and drifting (Zhang & Agrawala, 2025a); the latter preserve the geometric scene context well but suffer from visual artifacts in unobserved regions (Fig. S1).

## C IMPLEMENTATION DETAILS

### C.1 GENERAL SETUP

**Experiment Setup.** For evaluation, the dataset used in Section 4 is primarily collected from the online source Unsplash (by SQUARESPACE, 2013), supplemented with a small number of robotic

scenes from Open X-Embodiment (Collaboration et al., 2023). Example scenes are shown in Fig. S5. All experiments use Blender and Houdini as simulators. Depth maps required as method inputs are rendered directly from the simulator, and optical flow maps are computed using SEA-RAFT (Wang et al., 2024b).

For experiments in Table 1, each generated video contains 81 frames with a spatial resolution of $480 \times 832$. For experiments in Table 2, each video consists of 129, 177, or 225 frames—corresponding to 2, 3, and 4 temporal segments, respectively—with durations of 8s, 11s, and 14s, to reflect different difficulty levels.

**Backbone Models.** During video synthesis, we combine Image2V model, Wan2.2 (14B), Depth2V model, VideoX-Fun, and Flow2V model, VACE (Jiang et al., 2025), with the latter two built on Wan2.1 (14B) backbone.

For $f^{\text{camera}}$, we use the checkpoint from `https://huggingface.co/alibaba-pai/Wan2.1-Fun-V1.1-14B-Control-Camera`.

**Hyperparameters.** For SVGD module, We use the standard RBF kernel $k(x, x') = \exp(-\|x - x'\|^2/h)$ with heuristics bandwidth $h = \text{median}\{\|g(x^{(l)}) - g(x^{(l')})\|_2\}_{l \neq l'}$, and set $\eta = 1e - 3$ for Algorithm 1. For video generation, we use a resolution of 832x480, 81 frames (5 seconds with FPS of 16), inference step number as 30, and a classifier-free guidance scale of 6.0. For evaluation in Table 1 and Table S1, we set $L = 2$. For evaluation in Table 3 and Table S2, we set $L = 2, 4, 8$. The foreground mask $\mathcal{M}_{\text{fg}} \in \mathbb{R}^{H \times W \times N}$ is initially computed by duplicating the semantic segmentation mask for unconstrained foreground objects from input foreground image $\mathcal{I}_{\text{fg}}$ across $N$ frames. During the denoising process, $\mathcal{M}_{\text{fg}}$ is updated to be the predicted mask from a dynamic motion tracker Huang et al. (2025) for the currently generated video, once the unconstrained objects exhibit sufficiently clear dynamics for tracker to estimate. Based on the above intuition, we developed a two-stage mask adaptation strategy: during the high-noise denoising steps ($t = T, \ldots, T//2 - 1$), the initial mask is used. At $t = T//2$, $\mathcal{M}_{\text{fg}}$ is updated to be the predicted mask from the dynamic motion tracker.

**Score Composition and Sampling.** To compute the adaptive masked score in Eq. (6), we apply dilation and Gaussian blur to both $\mathcal{M}_{\text{fg}}$ and $\mathcal{M}_{\text{sim}}$ for smoother spatial transitions. Additionally, with background priors provided as context conditionals, we overlay the simulated depth or flow of foreground objects onto the estimated depth or flow of the background context. This alleviates the distribution shift for individual backbone models during sampling.

## C.2 IMAGE EDITING

We generate foreground-background first frame image pair $(\mathbf{I}_f, \mathbf{I}_b)$ using image editing techniques. Specifically, we generate the counterpart given either an input background image $\mathbf{I}_b$ or a foreground image $\mathbf{I}_f$.

For foreground image generation, we adopt the PoE framework (Zhang et al., 2025), where the 3D assets are inserted into the background image under the composition of fill, depth and canny-edge models (FLUX.1, 2024). The depth map and canny edges are extracted directly from the 3D asset to preserve geometric fidelity. To ensure realistic integration, we then apply a dense image harmonization model (Chen et al., 2023), which adjusts shading and lighting such that the inserted objects are consistent with the surrounding background.

For background image generation, we employ Qwen-Image-Edit (Wu et al., 2025) to remove the foreground objects in $\mathbf{I}_f$.

## C.3 COMPUTING CONTEXT CONDITIONALS VIA INVERSION

**Obtaining Reference Video.** We obtain the reference video $\mathcal{V}_{\text{bg}}$ in Section 3.2 as follows. First, $\hat{\mathcal{V}}_{\text{bg}}$ is generated by $f^{\text{camera}}$, producing 81 frames at a resolution of $832 \times 480$. This output is misaligned with $f^{\text{memory}}$, which only supports 121 frames at $704 \times 1280$. To align them, we resize $\hat{\mathcal{V}}_{\text{bg}}$ to match the resolution of $f^{\text{memory}}$ and apply RIFE (Huang et al., 2022b) for frame interpolation, expanding it to 121 frames. After re-rendering, we resize the video back to $832 \times 480$ and select 81 frames from

---

**Algorithm 2** Video Extension with Context Conditionals

---

1: **Inputs:** Number of video segments $S$; sequence of text prompts $\{\mathcal{Y}_s\}_{s=1}^S$; initial image $\mathcal{I}$; segment trajectories $\{\mathcal{C}_s\}_{s=1}^S$; pre-trained models $\{p^{(i)}(\cdot)\}_{i=1}^N$; annealing length $T$ and schedule $\{t_T, \cdots, t_0\}$.
2: **Init:** Set $\mathcal{V} \leftarrow \emptyset$
3: **for** $s = 1$ to $S$ **do**                                          ▷ Iterate over video segments
4:     **if** $s = 1$ **then**
5:         Initialize first frame with input image $\mathcal{I}$.
6:     **else**
7:         Use the last $K$ frames of $\mathcal{V}_{s-1}$ as overlap history.
8:         Set $\mathcal{M}_{\text{context}} = \mathbf{1}$ and assign context conditionals $z_t^{\text{context}}$ directly from $\mathcal{V}_{s-1}$ for these $K$ frames.
9:     **end if**
10:    Run the annealed update procedure (lines 4–14 of Algorithm 1) with $(\mathcal{Y}_s, \mathcal{C}_s)$ to synthesize segment $\mathcal{V}_s$.
11:    Concatenate $\mathcal{V} \leftarrow \mathcal{V} \,\|\, \mathcal{V}_s$.
12: **end for**
13: **Output:** Extended video $\mathcal{V}$ consisting of all segments with overlap ensuring temporal consistency.

---

the 121-frame sequence. The selection is based on MSE closeness between consecutive frames, removing frames with higher similarity until the sequence is reduced to 81 frames.

**Inversion.** The generated background $\mathcal{V}_{\text{bg}}$ provides a structural layout for video synthesis, ensuring that the simulator has an aligned camera trajectory for rendering. It also maintains 3D consistency across frames. During sampling, the background is preserved by inversion. We apply RF-Solver (Wang et al., 2024a) inversion on $\mathcal{V}_{\text{bg}}$. Inversion maps data back into noise, which reverses the sampling process. The inversion process of RF-Solver can be derived as:

$$\mathbf{z}_{t_{i+1}}^{\text{context}} = \mathbf{z}_{t_i}^{\text{context}} + (t_{i+1} - t_i)\, v_\theta(\mathbf{z}_{t_i}^{\text{context}}, t_i) + \frac{1}{2}(t_{i+1} - t_i)^2 v_\theta^{(1)}(\mathbf{z}_{t_i}^{\text{context}}, t_i), \qquad \text{(S8)}$$

where $\mathbf{z}_{t_i}^{\text{context}}$ and $\mathbf{z}_{t_{i+1}}^{\text{context}}$ denote the latents of the background context during inversion.

### C.4 FRAMEWORK INSTANTIATION FOR VIDEO EXTENSION

Our video extension implementation is detailed in Algorithm 2, where the variational inference component is omitted for simplicity. Comparative results with baselines are shown in Fig. S2. In our setup, each segment contains 81 frames with an overlap of $K = 33$ frames. For evaluation, we set varying video lengths of 129, 177, ad 255 frames—corresponding to 2, 3, and 4 segments, or durations of 8s, 11s, and 14s, respectively.

### C.5 AUTOMATIC EVALUATION WITH VLMS

We follow the protocol of PhysGen3D (Chen et al., 2025a) to run automatic evaluations for Section 4.1 and Section 4.2 using GPT-4o (Hurst et al., 2024). For each scene, 10 uniformly sampled frames from the outputs of all methods are sent to GPT-4o along with template prompts defined in PhysGen3D. GPT-4o assigns scores in the range of $[0, 1]$ for each video across all metrics. The scores for each metric are then averaged over all scenes.

## D  MORE EVALUATION RESULTS

We further scale our evaluation, with 8 camera trajectories ("move left/right", "push in", "pull out", "pan left/right", "orbit left/right") that represent most camera motion and 20 scenes, including indoor, outdoor, human, animals, vehicles, and robots. For simulation, our evaluation includes fluid dynamics, rigid body, and deformable objects. With combinations with these camera paths, scene types, and simulation categories, we extend the evaluation from 50 to 200 diverse and representative

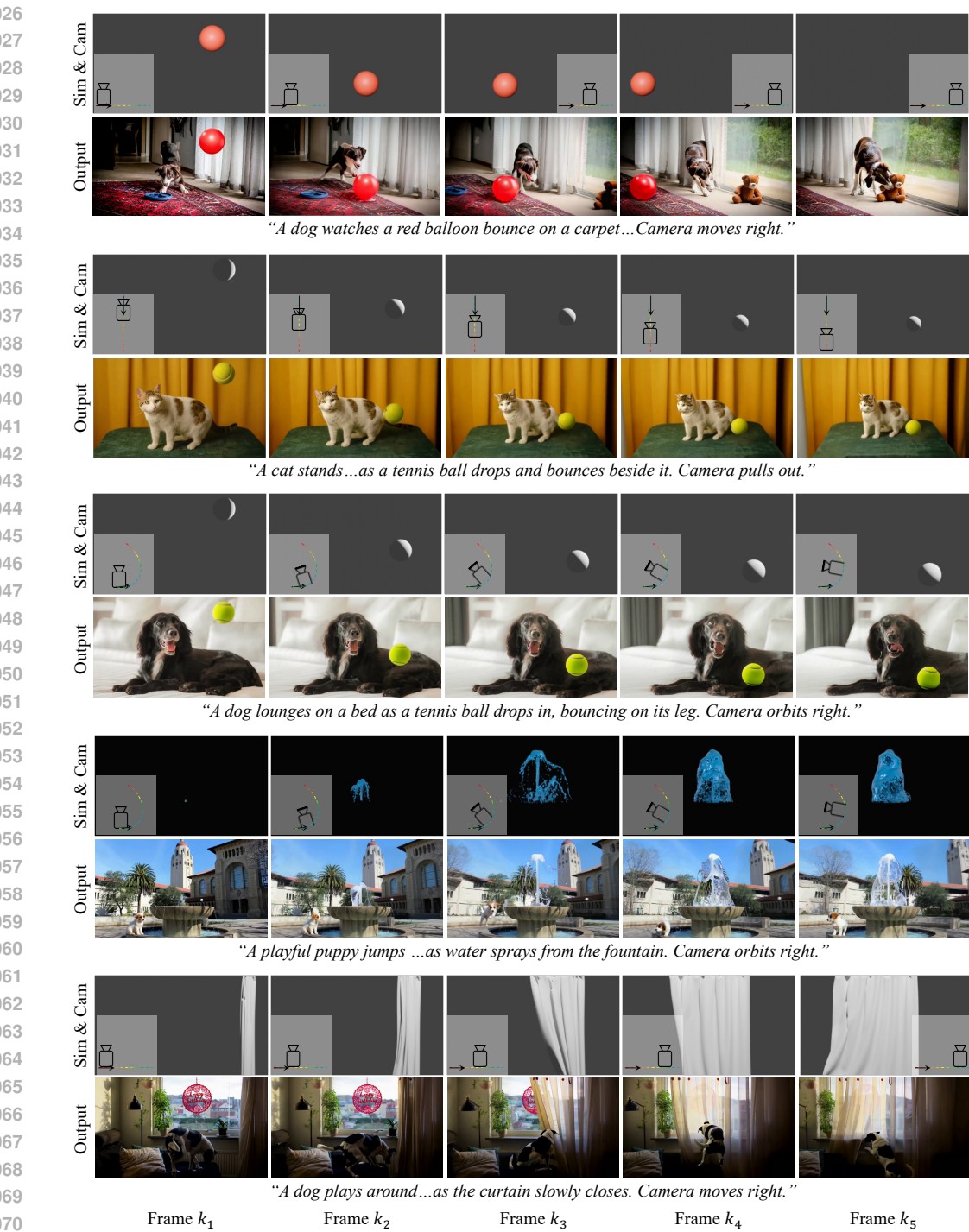

Figure S3: **More Qualitative Results.** For each test case, input object simulation and camera trajectory are visualized, with text prompts shown at the bottom. Our method generates videos aligned with object and camera trajectories while exhibiting natural and coherent content for unconstrained regions.

samples. The scaled-up evaluation results are shown in Table S1. We can observe the same trends in the 50-sample experiment in Table 1, demonstrating the robustness of our method.

| Methods | Controllability | | | Video Quality | | | | Semantic Alignment | |
|---|---|---|---|---|---|---|---|---|---|
| | LPIPS (↓) | Camera (↑) | GPT-4o (↑) | Subject (↑) | Photo (↑) | 3D Consist (↑) | GPT-4o (↑) | ViCLIP (↑) | GPT-4o (↑) |
| Image2V | 0.084±0.042 | 0.507±0.219 | 0.701±0.182 | 0.638±0.055 | 0.792±0.127 | 0.731±0.152 | 0.693±0.181 | 0.205±0.031 | 0.618±0.192 |
| Depth2V | 0.081±0.038 | 0.533±0.249 | 0.679±0.196 | 0.612±0.042 | **0.891**±0.081 | 0.821±0.083 | 0.688±0.182 | 0.202±0.028 | 0.598±0.187 |
| Flow2V | 0.071±0.034 | 0.427±0.277 | 0.689±0.178 | 0.618±0.039 | 0.789±0.123 | 0.731±0.178 | 0.692±0.174 | 0.218±0.026 | 0.599±0.169 |
| Cam2V | 0.091±0.045 | 0.653±0.221 | 0.661±0.205 | 0.623±0.037 | 0.769±0.112 | 0.757±0.101 | 0.677±0.081 | 0.201±0.034 | 0.562±0.176 |
| GEN3C | 0.098±0.044 | **0.848**±0.142 | 0.680±0.178 | 0.579±0.028 | 0.872±0.066 | 0.889±0.079 | 0.698±0.183 | 0.190±0.022 | 0.565±0.162 |
| PoE-I&D | 0.077±0.034 | 0.421±0.256 | 0.654±0.188 | 0.603±0.038 | 0.754±0.097 | 0.652±0.109 | 0.669±0.172 | 0.213±0.025 | 0.649±0.158 |
| PoE-C&D | 0.070±0.029 | 0.620±0.207 | 0.707±0.165 | 0.579±0.029 | 0.731±0.090 | 0.803±0.067 | 0.696±0.145 | 0.216±0.023 | 0.627±0.158 |
| Ours | **0.065**±0.022 | 0.783±0.168 | **0.781**±0.187 | **0.639**±0.031 | 0.795±0.079 | **0.902**±0.062 | **0.740**±0.161 | **0.223**±0.022 | **0.672**±0.150 |

Table S1: **Controllable Video Generation on Larger Evaluation Set.** We scale our evaluation to 200 test samples and the expanded experiment confirms the same trends observed in the 50-sample experiment in Table 1: 3D-based method GEN3C excels in 3D consistency but lags in other dimensions, while video generative models show better controllability and visual quality but weaker geometric consistency. Our method achieves better or comparable performance compared to these two classes of methods. These consistent trends across different evaluation sizes further demonstrate the robustness of our approach.

| Model | Clip Score↑ | RotErr↓ | TransErr↓ | CamMC↓ |
|---|---|---|---|---|
| FullDiT | 22.97 | 1.20 | 3.31 | 3.98 |
| Cam2V | 24.03 | 1.13 | 3.43 | 3.84 |
| Ours | **24.21** | **1.02** | **3.21** | **3.62** |

Table S2: **Quantitative Comparison of FullBench.** We compare our method with FullDiT, and Cam2V on camera-to-video generation. Our method achieves better text alignment and camera controllability.

Additionally, we evaluate our method on FullBench(Ju et al., 2025), a multi-task video generation benchmark. Among its various categories, the camera-to-video split is most relevant to our setting. This split contains 200 cases randomly selected from RealEstate10k (Zhou et al., 2018) test set. The evaluation employs 4 metrics across two key aspects: text alignment, and camera control. CLIP similarity (Radford et al., 2021) is used for text alignment. For camera control, we adopt RotErr, TransErr, and CamMC, as in CamI2V (Zheng et al., 2024). Results in Table S2 show that our method achieves better camera controllability and text-video alignment.

# E   MORE QUALITATIVE RESULTS

We present additional qualitative results in Fig. S3. In Fig. S5, we demonstrate our method on a robotic scene using the same background image prompt but different object simulations (e.g., the thin cloth on the cardboard box) and prompts for the robot arm. The resulting outputs reflect consistent and appropriate responses from the robot, showing that our method's ability to perform sim-to-real transfer.

# F   MORE ABLATIONS

We show an ablation on reconstruction loss in Fig. S4. Without reconstruction loss (w/o MSE), subtle discrepancies appear in the background compared to the background prior. While acceptable for controllable video synthesis—where the background need not exactly match the priors—for long video generation, it becomes critical. In particular, when the camera revisits previously seen viewpoints in later segments, reconstruction loss is essential for maintaining scene consistency. As shown in Table 2, removing the reconstruction loss leads to a notable drop in both photometric and 3D consistency.

Additionally, we report split results in the SVGD experiment in Table 3, where we evaluate $L \in 2, 4, 8$, corresponding to 2, 4, and 8 output videos, respectively. The detailed split results are provided in Table S3. We observe that the diversity gain remains consistent across different values of $L$ compared to baseline (w/o SVGD), with larger $L$ producing more diverse outputs.

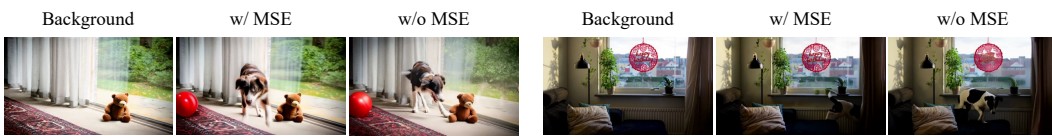

Figure S4: **Ablation on Reconstruction Loss.** Without reconstruction loss, subtle discrepancies may appear in the background compared to the provied background context.

| Method | SSIM↓ | LPIPS↑ | PSNR↓ |
|---|---|---|---|
| w/o SVGD | 0.754 | 0.279 | 20.947 |
| Ours (L = 2) | 0.746 | 0.286 | 20.602 |
| Ours (L = 4) | 0.742 | 0.297 | 20.314 |
| Ours (L = 8) | **0.739** | **0.305** | **20.178** |

Table S3: **Split Results for Output Diversity.** We report diversity metrics for each value of $L$ (2, 4, 8) separately. Consistent with Table 3, less similarity is better as it corresponds to higher diversity. Results show that larger $L$ yields lower similarity among particles, indicating higher output diversity.

## G  COMPUTATION REQUIREMENT

All experiments are conducted on a single NVIDIA H200 GPU. Generating one sample in Section 4.1 and one temporal segment in Section 4.2 takes approximately 40min. We also report the run-time, GPU memory, and GPUs used for each method in Table S4. As shown, the baselines used in our compositional framework are themselves highly memory- and time-intensive. However, our method is model-agnostic and can be applied to lighter and faster backbones, which would substantially reduce both runtime and memory usage.

## H  LIMITATIONS

Our method accepts four types of inputs: image prompts, text prompts, camera trajectories, and asset trajectories. Some modalities would introduce limitations:

**Image Domain Generalization.** Our approach may struggle with out-of-distribution (OOD) image inputs, such as robotic scenes or human hands, thin structures, and specular backgrounds, due to limited coverage in the training data. This makes precise control in such domains challenging.

**Fast Camera Roll.** If the camera rolls very fast, then the video model responsible for content creation might generate lower-quality, thus generating low quality conditional context.

**Asset Trajectory Complexity.** Highly dynamic or unnatural asset motions can pose difficulties, especially when they fall outside the training distribution.

We expect that task-specific finetuning of backbone models and larger-scale training on diverse assets and prompts will further mitigate these challenges.

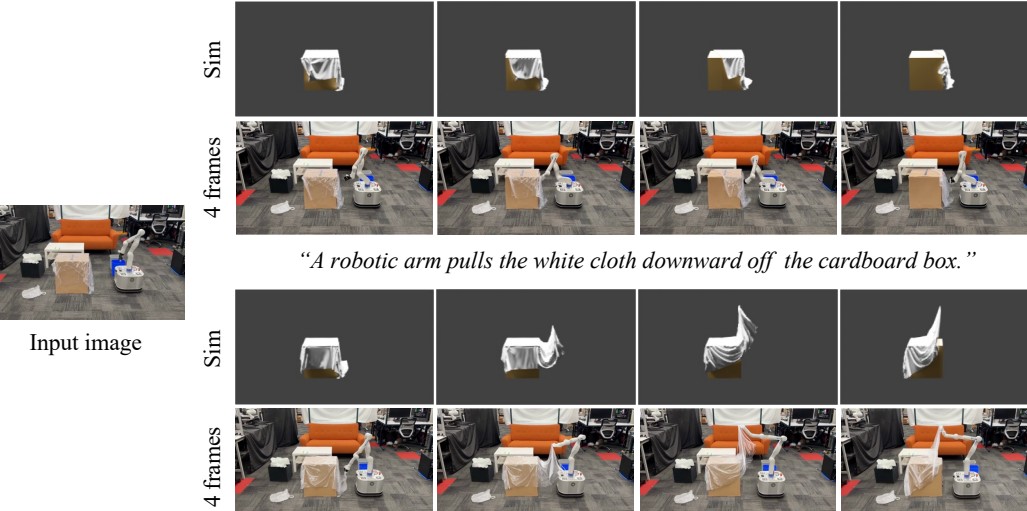

Figure S5: **Qualitative Result on Real-World Scene.** Given the same input image, different object simulations and text prompts lead to appropriate reactions from the unconstrained object (e.g., the robot arm). The input image is from TidyBot (Wu et al., 2023a).

| Method | Time (min) | Peak memory (GB) | GPUs |
|---|---|---|---|
| Image2V | 12 | 71 | 1 |
| Depth2V | 22 | 13 | 1 |
| Flow2V | 17 | 117 | 1 |
| Cam2V | 20 | 14 | 1 |
| GEN3C | 7 | 62 | 1 |
| PoE-I&D | 30 | 76 | 1 |
| PoE-C&D | 35 | 74 | 1 |
| Ours | 40 | 108 | 1 |
| Ours | 23 | 64 | 2 |

Table S4: **Comparison of Memory and Runtime with Baselines.** We report the runtime, peak GPU memory consumption, and number of GPUs required for each method. The baselines used in our compositional framework are themselves highly memory- and time-intensive. Our method remains competitive in efficiency and can generate a sample in 23 minutes when using 2 GPUs in parallel, while offering improved controllability and performance.

