# OpenReview forum: "Controllable Video Synthesis via Variational Inference"
_ICLR.cc/2026/Conference — Submitted to ICLR 2026_

### Official Review · Reviewer_c9w8 · 2025-10-18

**Soundness:** 3
**Presentation:** 3
**Contribution:** 2
**Rating:** 4
**Confidence:** 4

**Summary:**

The paper proposes an inference-time optimization framework built on top of a variational inference formulation. The method enables controllable video generation under one or multiple conditions such as text prompts, camera trajectories, and image inputs. It defines a product-of-experts target distribution that combines multiple pre-trained backbones (Wan2.2, Depth2V, VideoX-Fun, Flow2V, and VACE), each capturing a specific control modality. The optimization is performed using Stein Variational Gradient Descent (SVGD) to minimize the KL divergence between particle distributions and a sequence of annealed intermediate targets.

A key component of the approach is a 3D-aware context-conditioning mechanism, which anchors background and camera information to improve geometric consistency and mitigate local optima. This is implemented using external pretrained models such as Cam2V and GEN3C, together with an inversion process to compute context latents. Spatially adaptive masks are employed to balance the influence of different conditioning sources during optimization.

Although the method primarily represents an engineering effort, combining several pretrained systems and heuristic stabilizers, it is supported by a coherent theoretical interpretation through variational inference. The generated results demonstrate improved controllability, scene consistency, and visual quality compared to prior methods that rely on fixed conditioning. However, the computational cost is significant (approximately 40 minutes per video segment on an H200 GPU) making the approach resource-intensive.

**Strengths:**

- Strong controllability and flexibility: Supports multiple input modalities (text, image/video, camera, and simulation) in a unified generation process, which most existing methods cannot handle jointly.
- Conceptually interesting bridge: Combines heavy engineering composition (multiple pre-trained models) with a sound theoretical framing via variational inference and SVGD, giving the system a principled interpretation instead of a heuristic pipeline.
- Experimentally validated: Demonstrates clear performance gains in controllability and consistency, reaching or surpassing state-of-the-art results.

**Weaknesses:**

- Extremely high inference cost: Around 40 minutes per segment on a single H200 GPU makes the method impractical beyond research demos.
- Complex pipeline: Involves many external pre-trained components (Cam2V, GEN3C, RF-Solver, segmentation, inversion), increasing maintenance cost and dependency on each model’s quality.
- Bounded by backbone performance: While the method builds upon multiple pre-trained backbones, its optimization framework allows combining their complementary strengths, leading to performance exceeding individual baselines. However, it still remains conceptually bounded by the representational capacity of these experts.
- Limited originality in modeling: The novelty lies in aggregation and inference formulation, not in new architectures or learning strategies. A trainable unified model using these modalities directly would be a more scalable direction.
- Generalization uncertainty: The method inherits data biases from the underlying models, and may fail in domains not covered by them (e.g., robotics, human manipulation). This is a minor weakness, though every model has its own data biases, it is mentioned only for completeness.
- Missing compositional generation references: Related works should include recent object-centric and compositional methods such as Learning to Compose [1], SlotDiffusion [2], SlotAdapt [3], and SlotAdaptVideo [4] (not a necessity, but an unpublished extension of [3] to the video domain).
[1] Jung et al., Learning to Compose: Improving Object Centric Learning by Injecting Compositionality, ICLR 2024
[2] Wu et al., SlotDiffusion: Object-Centric Generative Modeling with Diffusion Models, NeurIPS 2023
[3] Akan & Yemez, Slot-Guided Adaptation of Pre-trained Diffusion Models for Object-Centric Learning and Compositional Generation, ICLR 2025
[4] Akan & Yemez, Compositional Video Synthesis by Temporal Object-Centric Learning, arXiv

**Questions:**

- I think one of the most important comparisons would be a runtime comparison across all methods. How does the proposed method perform relative to the baselines in terms of runtime?
- The reported 40-minute inference time per segment is substantial. Which part of the pipeline dominates the runtime, such as model evaluations, inversion for context latents, or SVGD iterations? Are there identifiable opportunities for reduction?
- The method relies on context conditionals derived from GEN3C outputs followed by inversion. Table 4 already includes results without context conditioning, but the question here concerns sensitivity to the quality of the reconstructed context rather than its removal. How sensitive is the overall optimization quality to errors or noise in this inversion process? In other words, does imperfect context reconstruction noticeably degrade consistency or controllability? Could the inversion be performed by an alternative method for comparison?
- Since the framework integrates multiple pre-trained backbones, it would be useful to understand how performance changes if certain experts are removed. Does the optimization remain stable with a reduced set of backbones, or is it strongly dependent on the full ensemble? This could be evaluated by removing one or more backbones during optimization.
- Could the proposed inference-time framework be extended to train a unified model capable of handling multi-modal conditioning directly, without iterative optimization? For example, can a unified model be trained using the proposed optimization scheme to eliminate the inference-time overhead?

---

> ### Author Response · Authors · 2025-11-25
> **Response 1/2**
>
> # Inference cost
> Thank you for raising this concern. Our current runtime relects the choice of backbone: we use wan-2.1 14B family, which is already extremely memory- and compute-intensive.
>
> However, our formulation is model-agnostic. It can be applied to other diffusion/flow-based models, including faster and lighter backbones. In practice, our method can also incorporate standard acceleration techniques, with more details deferred to the "Inference time reduction" section below in our response.
>
>
> # Complex pipeline and bounded by backbone performance
> Our pipeline uses several pretrained models, and this design is intentional and fundamental to our training-free compositional paradigm. Each pretrained model contributes a distinct capability: Cam2V provides camera control, GEN3C provides 3D consistency, etc. Our approach leverages the complementary strengths of these independently trained models. Any component can be swapped with a more efficient or higher-quality model as the field progresses, and there is no joint training or tight coupling, keeping maintainance cost low. This modular, plug-and-play design avoids the need to retrain massive unified models whenever a new capability or control modality is added.
>
>
> # Originality in modeling
> This work focuses on the sampling/inference aspect of video generation models, as opposed to proposing a new architecture or model training strategies.
>
> We further clarify that two critical advantages compared to trainable unified models:
> 1. **Reduced data curation cost**. Our method is training-free, reusing powerful pretrained video backbones. In contrast, unified controllable training paradigms such as FullDit require large curated datasets that contain all control modalities at various granularities: data that are extremely expensive or even infeasible to collect (e.g., camera poses, 4D asset, and real video triplets). With more conditional modalities involved, curating such data to have a full cover of the input distribution is increasingly costly due to the increasing dimensionality of input conditions.
> 2. **Inference flexibility**. A training-based approach cannot easily adapt to new or unseen control signals at inference time, while ours can naturally adapt to new and flexible controls by changing base model configurations.
>
>
> # Generalization uncertainty
> We agree with the reviwer that the method (and all data-driven methods) is limited on this front, and will include the discussion in the limitation section.
>
> # References
> We thank the reviewer for pointing out this important line of related works. We have included them in the revision in line 99.
>
> # Runtime
>
>
> We agree with the concern and have added more details below on the run-time, GPU memory:
>
>
> | Method          | Time (min) | Peak memory (GB) | GPUs |
> |-----------------|-----------:|-----------------:|-----:|
> | Image2V         |         12 |               71 |    1 |
> | Depth2V         |         22 |               13 |    1 |
> | Flow2V          |         17 |              117 |    1 |
> | Cam2V           |         20 |               14 |    1 |
> | GEN3C           |          7 |               62 |    1 |
> | PoE-I&D         |         30 |               76 |    1 |
> | PoE-C&D         |         35 |               74 |    1 |
> | Ours  |         40 |              108 |    1 |
> | Ours  |         23 |               64 |    2 |
>
> All experiments are conducted on H200 GPU(s). As shown, the baselines used in our compositional framework are themselves highly memory- and time-intensive. However, our method is model-agnostic and it can be applied to ligher and faster backbones, which would substantially reduce both runtime and memory usage.

---

> ### Author Response · Authors · 2025-11-25
> **Response 2/2**
>
> # Inference time reduction
>
> We thank the reviewer for raising this important question regarding computational efficiency. As we are using Wan2.1 14B family as pretrained backbones, the majority of the runtime is from the denoising process, which accounts for about 80% of the total runtime.
>
> While our method focuses on a training-free compositional formulation for controllable video synthesis and efficiency is orthogonal to our core contribution, our method is fully compatible with the acceleration techniques already used in industrial video generation systems. We discuss potential directions of improvements below.
> 1. **Distillation and caching techniques**. Because our method operates on standard diffusion/flow backbones, it can directly benefit from distillation, causal/auto-regressive training, TeaCache [1], and other latency-reduction methods, allowing substantial speedups without modifying our formulation.
> 2. **Architectural improvements**. Architecutral improvements including replacing separate base models with condition-specific LoRA adapters, as kindly pointed out by reviewer `WdWW`. Such architectural changes are fully compatible with our variational framework and will lead to much lower overhead both in latency and memory consumption.
> 3. **Parallelization**. Multi-GPU parallelization further improves latency, with empirical results on 2 GPUs instead of 1 in the original experiments included in the table above. We highlight that, for each annealing step, querying base models for score functions is embarassingly parallelizable **both across expert models and across particles** as they don't have dependency on each other, and it's straightforward to implement parallelism. These techniques, once integrated, make it feasible to scale up the proposed method.
>
> # Sensitivity to inversion process
> Imperfect context reconstruction does not noticeably degrade controllability. This is because the inversion process reliably preserves the spatial layout of the scene, and controllability for camera and object depends primarily on this layout rather than on precise pixel-level reconstruction.
>
> For consistency, there are two cases:
> 1. The camera does not revisit a previously seen viewpoint. In this setting, the background does not need to match the original context exactly. Minor appearance differences introduced by inversion do not affect scene coherence.
> 2. The camera revisits a past viewpoint (common in long-video generation). In this challenging scenario, inversion alone is insufficient to guarantee perfect consistency. As mentioned in Algorithm 1 line 11, we include an aditional reconstruction loss (MSE) to better preserve appearance. Qualitative examples are shown in Appendix Figure S4.
>
> We compare our RF-Solver-based inversion against DDIM inversion using MSE loss to quantify reconstruction accuracy:
>
> | Method          | MSE↓    |
> |-----------------|--------:|
> | DDIM            |  0.103  |
> | RF-Solver (Ours)|  **0.055**  |
>
> As shown, our method has better reconstruction accuracy than DDIM inversion method.
>
>
>
> # Performance changes with experts removed
> In our pipeline, when two main pretrained backbones form the core of the compositional process, removing one of them effectively reduces our approach to a single-model baseline, which is what we report in Table 1 of the paper. Conceptually, removing a backbone disables the specific capability it contributes. For example, if the Cam2V backbone is removed while Depth2V remains, camera controllability naturally degrades, whereas object dynamics guided by depth remain comparable.
>
> # Extension to remove iterative optimization
> Thank you for this thoughtful question. Yes, the proposed framework can indeed support training a unified multi-controllable model. Our method can be used to generated synthetic paired data that include diverse combinations of controls (text, camera trajectories, object trajectories, etc.), which are otherwise extremely difficult to obtain. Such synthetic supervision can then be used to train a single unified model that directly handles multi-conditioning without iterative optimization and thus eliminate the inference-time overhead. This is one of the main contributions, and could potentially be a promising extension of our work.
>
> > [1] Timestep Embedding Tells: It's Time to Cache for Video Diffusion Model. Feng Liu, Shiwei Zhang, Xiaofeng Wang, Yujie Wei, Haonan Qiu, Yuzhong Zhao, Yingya Zhang, Qixiang Ye, Fang Wan. CVPR 2025.

---

> > ### Comment · Reviewer_c9w8 · 2025-11-27
> > **Post-rebuttal comments**
> >
> > I thank the authors for the detailed and concrete response, including additional tables and clarifications. Below I comment on the main points that relate to my original concerns.
> >
> > - **Inference cost and runtime table:** The new runtime and memory table is helpful. It clarifies that the proposed method is slower and more memory-hungry than any single backbone, and that much of this cost comes from the choice of Wan2.1 14B backbones and from the denoising process itself, which the authors state accounts for about 80 percent of the runtime. I agree that the formulation is in principle model-agnostic and could benefit from lighter backbones or standard acceleration techniques. However, in the current empirical form the method remains expensive, and this still counts as a practical limitation. I appreciate that the authors now present the cost comparison explicitly.
> >
> > - **Complex pipeline and bounded by backbone performance:** The clarification that the pipeline is intentionally modular, with each pretrained model contributing a distinct capability, alleviates some of my concern about maintainability. The plug-and-play argument, where components can be swapped as better backbones appear, is reasonable. At the same time, the method does inherit both the computational footprint and the biases of these backbones, so my original point about practicality and dependence on existing models still holds, although it is now better contextualized.
> >
> > - **Originality and position relative to unified models:** The explanation about reduced data curation cost and increased inference-time flexibility compared to unified multi-condition training (for example FullDiT) is convincing and clarifies the motivation for focusing on a training-free compositional approach. I still view the main novelty as lying in the inference formulation and system design rather than in new modeling components, but the rebuttal makes the design choices more clearly justified.
> >
> > - **Generalization uncertainty:** The authors acknowledge the limitations on generalization and plan to include this in the limitations section. This matches my intention, which was to flag this as a minor but real issue rather than a central flaw.
> >
> > - **References to compositional and object-centric work:**
> > The added references to compositional generation and related object-centric work address my comment on missing citations.
> >
> > - **Sensitivity to inversion and context quality:** The extended discussion of RF-Solver versus DDIM inversion, including the MSE comparison, directly addresses my question about sensitivity to context quality. It would strengthen the paper if this quantitative comparison and the two regimes (revisit versus non-revisit viewpoints) were summarized in the main text, since they are important for understanding robustness.
> >
> > - **Effect of removing experts:** The clarification that removing one of the two main backbones effectively reduces the method to a single-model baseline, as reported in Table 1, is reasonable. A short explicit remark to this effect in the paper would help readers who have the same question.
> >
> > - **Extension to a unified model:** I appreciate the discussion that the framework could be used to generate synthetic paired data to train a unified multi-controllable model in future work. This aligns well with my original suggestion and clarifies how the current method could be leveraged to reduce inference-time cost in a subsequent stage.
> >
> > - **Additional question on timestep-distilled backbones:** Given that most of the runtime arises from denoising with Wan2.1 14B, it would be useful to understand how the method interacts with timestep-distilled or otherwise step-reduced variants of these backbones. The rebuttal mentions that the framework is compatible with distillation and caching techniques such as TeaCache. Have the authors considered or tested the framework with a timestep-distilled Wan2.1 14B model (or an equivalent reduced-step backbone), and if so, how does this affect both runtime and controllability or consistency metrics? It would be helpful to know whether the proposed SVGD-based optimization behaves similarly when the underlying diffusion process has been shortened, or whether there are noticeable trade-offs between step reduction and compositional performance.
> >
> > Overall, the rebuttal improves the clarity and supports the soundness of the approach. My overall assessment remains that this is a solid and well-executed inference-time framework with strong controllability, but with significant computational cost and limited modeling originality.

---

> ### Author Response · Authors · 2025-12-03
>
> We sincerely thank the reviewer for the detailed and thoughtful comments. We are glad to see that our clarifications regarding inference cost, generalization, and inversion sensitivity were helpful and improvide the clarity of our approach. Below we respond to the final open points and offer a general reflection on contributions of our framework.
>
> ## Inference cost
> We agree that the current instatiation using Wan2.1 14B backbones is memory- and time-intensive. Our goal in using these large backbones was to demonstrate the full potential of controllable video synthesis at high visual quality. Our method produces significantly higher-quality videos than existing models (including those MCMC-sampling based PoE or unified training baselines FullDit), and uniquely supports multiple fine-grained user controls at once (e.g., simultaneous camera + object + text input), which prior models are not designed to handle. We view this as a critical step forward in controllability and sample fidelity. That said, our framework is fully compatible with acceleration strategies, and we will expand this discussion in the final version and include concrete results.
>
> ## Pipeline Complexity and Backbone Dependence
> We appreciate the reviewer’s balanced perspective. In our experiments, there existed combinations of multiple pretrained models, such as Cam2V, Depth2V, Flow2V, Image2V, and so on, all of which yielded strong performance, highlighting the robustness of our method. The training-free paradigm is indeed bounded by backbone performance, but it also benefits from their strengths and allows for continual improvement as more pretrained models emerge. Furthermore, the flexibility to flexibly substitute lighter and faster models (e.g., small diffusion or flow models with LoRA adapters) ensures the long-term adaptability and scalability of our approach.
>
> ## Originality and position relative to unified models
>
> While our approach builds on established inference techniques, its core contribution is not the introduction of a new component, but rather the design of a scalable, training-free framework for multi-controllable video synthesis. The key novelty lies in formulating a variational inference-based, multi-constraint optimization framework that enables flexible and compositional integration of diverse user controls, including text, camera trajectories, and object motion. To our knowledge, no prior work supports high-quality video generation with this level of control and compositionality. Our method not only achieves stronger controllability but also consistently matches or exceeds prior models in quality and 3D consistency, even compared to models trained for specific control settings. This highlights the robustness and generality of our framework.
>
> ## Additional question on timestep-distilled backbones
> Thank you for this thoughtful and constructive question. While we have not yet tested our method with a timestep-distilled variant of Wan2.1 since such a model is currently unavailable, our framework is fully compatible with acceleration strategies of this kind and ca readily incorporate them once they become available. We expect that using timestep-reduced models would substantially reduce runtime, especially considering that over 80% of our computational cost stems from denoising. That said, shorter diffusion trajectories may reduce sample diversity or controllability in some cases. However, since SVGD maintains multiple particles and adapts gradients during sampling, it can partially compensate for this reduction in depth.

---

### Official Review · Reviewer_dzZe · 2025-10-30

**Soundness:** 3
**Presentation:** 3
**Contribution:** 2
**Rating:** 4
**Confidence:** 4

**Summary:**

The paper aims to extend the capabilities of video generative models during inference time by steering generation outputs to faithfully follow a range of user control signals. The authors formulate the task as variational inference to approximate a composed distribution, leveraging multiple video generation backbones to collectively satisfy all task constraints. The paper presents compelling qualitative results showing videos that successfully follow various control inputs.

**Strengths:**

- Addresses an important and practical problem of utilizing multiple expert models for controllable video synthesis.
- The paper is well written and easy to follow.
- The Appendix provides strong qualitative support, including diverse and convincing video examples demonstrating the proposed method’s effectiveness.

**Weaknesses:**

1. **Incremental contribution**: The paper's novelty is incremental, not fundamental. It primarily applies the existing SVGD framework to a new task, conceptually mirroring prior variational-inference-based diffusion works [1, 2]. The annealed sampling is seen as an engineering heuristic, not a core algorithmic contribution. The work is a successful application of established methods.

2. **Computation resources analysis**: Please provide computational usage comparison against other baselines. The paper requires a detailed comparison of runtime, computational cost against established baselines, particularly the MCMC-based samplers.

3. **Limited experimental scope on long video generation**: To credibly demonstrate the method's scalability for long video generation, the paper must include comparisons against state-of-the-art long video generation sampling methods (e.g.,[3], [4]) on standardized benchmarks. The current evaluation feels more like an interesting, extended application of the proposed method than a systematic assessment. Also, please provide the qualitative video examples of generated long videos.

4. **Diversity concern**: The diversity gains attributed to SVGD are quantitatively marginal. The role of the particle count ($L$) in the experiments and evaluation is unclear. It appears the study uses $L=2$, but utilizing such a small particle count is ambiguous and concerning as it carries a high risk of mode collapse.

[1] Particle Guidance: non-I.I.D. Diverse Sampling with Diffusion Models, Corso et al., ICLR 2024\
[2] Collaborative Score Distillation for Consistent Visual Editing, Kim et al., NeurIPS 2023\
[3] FIFO-Diffusion: Generating Infinite Videos from Text without Training, Kim et al., NeurIPS 2024\
[4] DiTCtrl: Exploring Attention Control in Multi-Modal Diffusion Transformer for Tuning-Free Multi-Prompt Longer Video Generation, Cai et al., CVPR 2025

**Questions:**

Please refer to the Weakness Section. I am willing to increase my scores if my concerns are fully addressed.

---

> ### Author Response · Authors · 2025-11-25
>
> We thank the reviewer for the very constructive feedback. Please find our response below.
>
> # 1. Contribution
> We would like to highlight that, the contribution of this work focus on the practical application for multi-conditioned controllable video synthesis problem. Our setting involves composing a mixture of user controls (text, image/video, camera/asset trajectories) at inference time, a domain where no prior work provides a general, training-free solution.
>
> To solve this problem, we build on the well-established principle of variational inference and use SVGD as one of the core techniques, as discussed in the related works (lines 106 - 131). However, directly adapting such techniques to our problem setup is infeasible due to the high dimensionality and cost of video model sampling.
>
> This necessitates the introduction of 3D-aware context conditioning and factorization techniques proposed in this work. The role of these technical components, as well as the overall performance of our method, are extensively evaluated in experiments. You can see from Table 1 (50-sample evaluation) and Table S1 (200-sample evaluation) that our method achieves better or comparable 3D consistency, controllability, and visual quality at the same time, whereas baselines perform well only in one of these aspects.
>
> # 2. Computation resources
> Here is the computational usage comparison along with runtime against other baselines:
>
> | Method          | Time (min) | Peak memory (GB) | GPUs |
> |-----------------|-----------:|-----------------:|-----:|
> | Image2V         |         12 |               71 |    1 |
> | Depth2V         |         22 |               13 |    1 |
> | Flow2V          |         17 |              117 |    1 |
> | Cam2V           |         20 |               14 |    1 |
> | GEN3C           |          7 |               62 |    1 |
> | PoE-I&D         |         30 |               76 |    1 |
> | PoE-C&D         |         35 |               74 |    1 |
> | Ours  |         40 |              108 |    1 |
> | Ours  |         23 |               64 |    2 |
>
> All experiments are conducted on H200 GPU(s), where PoE-I&D and PoE-C&D are methods using MCMC-based sampling. As shown, the baselines used in our compositional framework are themselves highly memory- and time-intensive. However, ours is model-agnostic and it can be applied to ligher and faster backbones, which would substantially reduce both runtime and memory usage.
>
> # 3. Scope on long video generation
> We thank the reviewer for pointing this out. To the best of our knowledge, there are currently no long video generation methods or benchmarks suitable for our multi-controllable setting. Existing methods (e.g., [3], [4]) are either image to video generation or text to video generation without a mixture of user controls. We have provided qualitative video examples in Appendix Figure S2. Our method can effectively preserve contextual details from earlier segments. For example, it retains the scenery outside the window and preserves the decorative window patterns after the curtain closes and reopens (left panel of Figure S2). In addition, the method maintains scene consistency when the camera returns to previously visited scenes (right panel of Figure S2).
>
> # 4. Diversity
> We apologize for the ambiguity. In the main evaluation (Table 1), we use L=2. For the SVGD experiment (Table 3), our evaluation contains L=2, L=4, L=8, which results in 2, 4, and 8 output videos, respectively. The split results are reported below:
>
> | Method             |  SSIM↓     |  LPIPS↑      |  PSNR↓      |
> |--------------------|-----------:|-------------:|------------:|
> | w/o SVGD           | 0.754  | 0.279        | 20.947  |
> | Ours (L = 2)    | 0.746 | 0.286 | 20.602 |
> | Ours (L = 4)    | 0.742      | 0.297       | 20.314      |
> | Ours (L = 8)    | **0.739**      | **0.305**    | **20.178**      |
>
> We can see that the diversity gain remains consistent across different values of $L$, with larger $L$ producing more diverse outputs.

---

### Official Review · Reviewer_WdWW · 2025-11-01

**Soundness:** 3
**Presentation:** 3
**Contribution:** 3
**Rating:** 6
**Confidence:** 4

**Summary:**

This paper proposes a variational inference framework for controllable video synthesis that unifies multiple user controls (text, image, camera, and 4D asset trajectories). The method composes several pretrained video models into a single target distribution and optimizes it via annealed Stein Variational Gradient Descent (SVGD). A context-conditioned factorization improves 3D consistency and mitigates local optima. Experiments show clear gains in controllability, visual fidelity, and diversity over baselines like Wan2.2, GEN3C, and PoE.

**Strengths:**

### Originality
1. Compared with prior approaches that train separate models for different control conditions, this paper introduces a novel variational-inference framework for controllable video generation, which integrates multiple pretrained models into a unified and coherent compositional generative process.

### Technical Quality
1. The algorithmic design is rigorous: equations (1)–(6) are mathematically consistent and clearly linked to the pseudocode in Algorithm 1. Moreover, the paper provides a concrete framework that implements these theoretical derivations in practice, achieving a unified and controllable video generation model under the variational inference paradigm.
2. The method is implemented carefully with detailed ablations, quantitative evaluations (LPIPS, 4 metrics from WorldScore, ViCLIP, GPT-4o scores), and qualitative comparisons.

### Clarity
1. The exposition is clear and well-structured.
2. Implementation details and reproducibility statements are well documented in the appendix.

### Significance
1. The work addresses an important and timely challenge: combining multiple control modalities for coherent and physically consistent video generation.
2. The proposed formulation is general, training-free and can be applied on top of any modern diffusion or flow-based video model.

**Weaknesses:**

1. Computational and Memory Cost, and Scalability.
While the paper claims that the proposed method enables efficient sampling, this efficiency should be supported by fair comparisons and quantitative evidence. The discussion of cost should not be limited to computational complexity — memory consumption also deserves attention. The paper should report more detailed runtime or memory statistics beyond the brief mention in Appendix F.

2. Lack of Clear Motivation for Variational Inference.
The paper does not sufficiently explain why variational inference is preferable to the first category of methods mentioned in the Introduction — i.e., training a controllable video model directly on curated, multi-condition datasets. The authors should make explicit the advantages of adopting the variational-inference approach (e.g., reusability of pretrained models, reduced data curation cost) and clarify why fine-tuning a unified controllable model is not an adequate alternative (e.g. Full-DiT[1]).

[1] Ju X, Ye W, Liu Q, et al. Fulldit: Multi-task video generative foundation model with full attention[J]. arXiv preprint arXiv:2503.19907, 2025.

**Questions:**

My biggest concern is efficiency, and whether it can support industrial applications (because video generation is a matter of concern in the industry, and there are already some industrial applications).

1. If the method indeed incurs substantial computational and memory costs due to maintaining multiple particles and combining several large backbone models, would this make it difficult to deploy or scale in industrial applications?

2. In terms of computational efficiency, could the framework be accelerated using state-of-the-art distillation techniques as adopted in existing methods? For memory efficiency, instead of combining multiple distinct base models, is it possible to use a single backbone model with multiple condition-specific LoRA adapters, achieving similar compositional control with lower memory overhead?

---

> ### Author Response · Authors · 2025-11-25
>
> We appreciate your insightful comments and suggestions. We provide responses and clarifications below.
>
> # 1. Runtime and memory statistics
>
> We agree with the concern and have added more details below on the run-time, and GPU memory:
>
>
> | Method          | Time (min) | Peak memory (GB) | GPUs |
> |-----------------|-----------:|-----------------:|-----:|
> | Image2V         |         12 |               71 |    1 |
> | Depth2V         |         22 |               13 |    1 |
> | Flow2V          |         17 |              117 |    1 |
> | Cam2V           |         20 |               14 |    1 |
> | GEN3C           |          7 |               62 |    1 |
> | PoE-I&D         |         30 |               76 |    1 |
> | PoE-C&D         |         35 |               74 |    1 |
> | Ours  |         40 |              108 |    1 |
> | Ours  |         23 |               64 |    2 |
>
> All experiments are conducted on H200 GPU(s). Please refer to the last section for techniques to improve computation and memory cost. As shown, the baselines used in our compositional framework are themselves highly memory- and time-intensive. However, ours is model-agnostic and it can be applied to ligher and faster backbones, which would substantially reduce both runtime and memory usage. We have included this discussion in our revised Appendix G.
>
>
> # 2. Motivation for Variational Inference.
>
>
> We appreciate the reviewer’s question. The motivation of variational inference is that it enables diverse yet constraint-consistent video samples adhering to a mixture of user controls at inference time. Specifically, coarse controls such as text retain diversity (e.g., “man is panicked” but can be of different motion patterns), while precise controls such as camera/object trajectories remain consistent across samples. In practice, our approach produces significantly more diverse samples than prior compositional methods, such as [1].
>
> We explicitly list out two main advantages below:
> 1. **Reduced data curation cost**. Our method is training-free, reusing powerful pretrained video backbones. In contrast, unified controllable training paradigms such as FullDit require large curated datasets that contain all control modalities at various granularities: data that are extremely expensive or even infeasible to collect (e.g., camera poses, 4D asset, and real video triplets). With more conditional modalities involved, curating such data to have a full cover of the input distribution is increasingly costly due to the increasing dimensionality of input conditions.
> 2. **Inference flexibility**. A training-based approach cannot easily adapt to new or unseen control signals at inference time, while ours can naturally adapt to new controls by changing base model configurations.
>
>
> # 3. Improving latency and memory footprint
> We thank the reviewer for raising this important question and several key directions for improvement.
>
> While our method focuses on a training-free compositional formulation for controllable video synthesis and efficiency is orthogonal to the core contribution, our method is fully compatible with the acceleration techniques already used in industrial video generation systems. We discuss potential directions of improvements below.
> 1. **Distillation and caching techniques**. Because our method operates on standard diffusion/flow backbones, it can directly benefit from distillation, causal/auto-regressive training, TeaCache [2], and other latency-reduction methods, allowing substantial speedups without modifying our formulation.
> 2. **Architectural improvements**. Yes, it is indeed possible to use condition-specific LoRA adapters. Such architectural changes are fully compatible with our variational framework and will lead to much lower overhead both in latency and memory consumption.
> 3. **Parallelization**. Multi-GPU parallelization further improves latency, with empirical results on 2 GPUs instead of 1 in the original experiments included in the table above. We highlight that, for each annealing step, querying base models for score functions is parallelizable **both across expert models and across particles** as they don't have dependency on each other, and it's straightforward to implement parallelism. These techniques, once integrated, make it feasible to scale up the proposed method.
>
>
>
> > [1] Product of Experts for Visual Generation. Yunzhi Zhang, Carson Murtuza-Lanier, Zizhang Li, Yilun Du, Jiajun Wu. arXiv:2506.08894, 2025.
>
> > [2] Timestep Embedding Tells: It's Time to Cache for Video Diffusion Model. Feng Liu, Shiwei Zhang, Xiaofeng Wang, Yujie Wei, Haonan Qiu, Yuzhong Zhao, Yingya Zhang, Qixiang Ye, Fang Wan. CVPR 2025.

---

### Official Review · Reviewer_fepN · 2025-11-04

**Soundness:** 3
**Presentation:** 2
**Contribution:** 3
**Rating:** 6
**Confidence:** 2

**Summary:**

The paper proposes a variational-inference framework for controllable video synthesis that composes multiple pretrained video generators via a product-of-experts objective. The author propose "3D‑aware context conditioning"  to  generate “context conditionals” at every step to stabilize geometry and camera control. Experiments report stronger controllability and 3D consistency than baselines.

**Strengths:**

1. Casting multi‑constraint control as approximating a PoE target, then solving it with annealed forward‑KL and SVGD, is coherent and computationally attractive relative to MCMC‑based PoE sampling in prior work.
2. Extensive experiments are conducted with multiple settings and ablation studies.

**Weaknesses:**

## Major Concerns
1. The main evaluation uses only 50 test samples constructed from 10 simulations, 8 camera trajectories, 13 scenes, with 2 samples per method (Sec. 4.1). While the setups are well‑controlled, this feels small for claims of generality.
2. The proposed method aggregates multiple heavy backbones at inference. The wall-time, GPU usage should be also compared with baseline models and in ablation studies.
3. The GAP between factorization in Eq3-5 to the score in Eq6 is huge. Can the author provide some intuitive explaination, despite there are derivations in the appendix.
4. The adaptive foreground mask starts from image segmentation and is later replaced by a dynamic motion tracker once dynamics are sufficiently clear. What "sufficiently clear" is not clearly specified.

## Minor Concerns
5. There is '4D asset' in line 84, and '3D asset' in line 124. I think they all should be '4D'?
6. The Figure 2 is not clear to me. This figure should show a more clear workflow of the pipeline.
7. In the appendix, many \cite should be \citep. For example, in line 898.
8. Reproductibility: In the implementation details part, it is better to provide the detailed hyperparameter chosen.
9. Line 364: Cam2C should be Cam2V.

**Questions:**

Can you share typical failures (e.g., fast camera roll, thin structures, specular backgrounds) and how they relate to the context conditional term?

---

> ### Author Response · Authors · 2025-11-25
> **Response 1/2**
>
> We thank the reviewer for the constructive and detailed feedback.
> # 1. Larger evaluation set
> We agree that expanding the evaluation is valuable. While our multi-controllable setting is unique and no existing dataset fully matches the multi-constraint setup required in our experiments, we have nevertheless scaled our evaluation to 200 test samples, with 8 camera trajectories (“move left/right”, “push in”, “pull out”, “pan left/right”, “orbit left/right”) that represent most camera motion and 20 scenes, including indoor, outdoor, human, animals, vehicles, and robots. For simulation, our evaluation includes fluid dynamics, rigid body, and deformable objects. With combinations with these camera paths, scene types, and simulation categories, we extend the evaluation from 50 to 200 diverse and representative samples. The updated evaluation results are shown below:
>
> | Method          | LPIPS | Camera | GPT-4o | Subject      | Photo      | 3D Consist   | GPT-4o     | ViCLIP      | GPT-4o     |
> |-----------------|------:|-------:|-------:|-------------:|-----------:|-------------:|-----------:|------------:|-----------:|
> | Image2V         | 0.084 | 0.507  | 0.701  | 0.638 | 0.792      | 0.731        | 0.693      | 0.205       | 0.618      |
> | Depth2V         | 0.081 | 0.533  | 0.679  | 0.612        | **0.891**  | 0.821        | 0.688      | 0.202       | 0.598      |
> | Flow2V          | 0.071 | 0.427  | 0.689  | 0.618        | 0.789      | 0.731        | 0.692      | 0.218 | 0.599      |
> | Cam2V           | 0.091 | 0.653  | 0.661  | 0.623        | 0.769      | 0.757        | 0.677      | 0.201       | 0.562      |
> | GEN3C           | 0.098 | **0.848** | 0.680 | 0.579        | 0.872 | 0.889 | 0.698 | 0.190       | 0.565      |
> | PoE-I&D         | 0.077 | 0.421  | 0.654  | 0.603        | 0.754      | 0.652        | 0.669      | 0.213       | 0.649 |
> | PoE-C&D         | 0.070 | 0.620  | 0.707 | 0.579        | 0.731      | 0.803        | 0.696      | 0.216       | 0.627      |
> | Ours  | **0.065** | 0.783 | **0.781** | **0.639**   | 0.795      | **0.902**   | **0.740**   | **0.223**   | **0.672**   |
>
>
> The expanded experiment confirms the same trends observed in the original experiment: 3D-based method GEN3C excels in 3D consistency but lags in other dimensions, while video generative models show better controllability and visual quality but weaker geometric consistency. Our method achieves better or comparable performance compared to these two classes of methods.
>
> Additionally, we evaluate our method on standard benchmark from FullDiT [1]:
>
>
> | Model          | Clip Score↑| RotErr↓ | TransErr↓ |   CamMC↓  |
> |----------------|-----------:|--------:|----------:|----------:|
> | FullDiT        |     22.97  |    1.20 |    3.31   | 3.98      |
> | Cam2V          |     24.03  |    1.13 |    3.43   | 3.84      |
> | Ours |     **24.21**  |    **1.02** |    **3.21**   | **3.62**      |
>
> The results show that our method achieves better camera controllability and text-video alignment.
>
> # 2. GPU usage
>
> We agree with the concern and have added detailed experiments. The table below reports the run-time, GPU memory, and GPUs used for each method:
>
>
> | Method          | Time (min) | Peak memory (GB) | GPUs |
> |-----------------|-----------:|-----------------:|-----:|
> | Image2V         |         12 |               71 |    1 |
> | Depth2V         |         22 |               13 |    1 |
> | Flow2V          |         17 |              117 |    1 |
> | Cam2V           |         20 |               14 |    1 |
> | GEN3C           |          7 |               62 |    1 |
> | PoE-I&D         |         30 |               76 |    1 |
> | PoE-C&D         |         35 |               74 |    1 |
> | Ours  |         40 |              108 |    1 |
> | Ours   |         23 |               64 |    2 |
>
> All experiments are conducted on H200 GPU(s). As shown, the baselines used in our compositional framework are themselves highly memory- and time-intensive. However, our method is model-agnostic and can be applied to lighter and faster backbones, which would substantially reduce both runtime and memory usage.
>
>
> # 3. Clarifications on equations
> Thank you for pointing it out and please find more clarifications here. Eqns. (3-5) define our factorized distribution using the context conditionals $z^{\star}$ and Eqn. (6) is its concrete implementation. In Eqn. (6), we instantiate this choice as the context latent at timestep t, i.e., $z^{context}_t$, and both model terms $f^{image}$ and $f^{\psi}$ are conditioned on this context conditionals $z^{context}_t$, exactly mirroring how each target distribution $p^{(i)}(x)$ in Eqn. (4) is conditioned on $z^{\star}$.
>
> > [1] Fulldit: Multi-task video generative foundation model with full attention. Xuan Ju, Weicai Ye, Quande Liu, Qiulin Wang, Xintao Wang, Pengfei Wan, Di Zhang, Kun Gai, Qiang Xu. arXiv preprint arXiv:2503.19907, 2025

---

> ### Author Response · Authors · 2025-11-25
> **Response 2/2**
>
> # 4. Clarifications on mask adaptation
> We thank the reviewer for pointing out the ambiguity in the original phrases. We now clarify this by first outlining the observations and then providing a concrete description of the mask adaptation.
>
> We observed that in the early stages of denoising, the latents contain noise and the dynamic motion is ambiguous. After a certain number of denoising iterations (typically ~50% of the denoising schedule), object motion becomes distinguishable enough for the motion tracker to estimate a reliable dynamic foreground mask.
>
> Based on the above intuition, we developed a mask adaptation strategy: during the high-noise denoising steps (for t=T, ..., T//2-1), the foreground mask $M_{fg}$ is computed by duplicating the semantic segmentation mask for unconstrained foreground objects from input foreground image across N frames. When t=T//2, $M_{fg}$ is updated to be the predicted mask from a dynamic motion tracker [1].
>
> # Other clarifications
> 5. “4D asset” vs. “3D asset and their pose sequences”. These two terms are equal.
> 6. Figure 2 clarity. **Top-left:** Inputs, including text prompt, camera trajectory, foreground and background images, and 3D asset trajectory from simulator. **Bottom-left:** we then compute context conditionals. **Bottom-center:** Demonstration of our denoising process, where each row visualizes the latent at a certain timestep. The latent is composed of 3 areas, context conditionals (gray area), foreground objects (green area), and simulated assets (orange area). **Right:** The final generated video.
> 8. Many \cite should be \citep. Thank you, we have fixed it.
> 9. Hyperparameters for reproducibility. We expanded the implementation details section with full hyperparameters chosen in line 884. We will release full code upon acceptance.
> 10. “Cam2C” -> “Cam2V” typo. Thank you! Corrected.
>
> # Failure modes
> We thank the reviewer for pointing out the two important modes of failures, which we expand below.
>
> The first is fast camera roll. Our method is 3D-aware and can handle complex camera trajectories (e.g., orbit, push-in). However, if the camera rolls very fast, then the video model responsible for content creation might generate lower-quality frames because such viewpoints are uncommon in the pretraining data, thus generating low quality conditional context even if the camera control is accurate.
>
> The second is visual artifacts such as thin structure, or specular backgrounds often lie outside the distribution of the pretrained models we rely on. Our approach is training-free and inherits the limitations of these backbones. With that said, the quality of our method will also improve with better backbones. We also clarify in the limitation section in line 1081.
>
> > [1] Segment Any Motion in Videos. Nan Huang, Wenzhao Zheng, Chenfeng Xu, Kurt Keutzer, Shanghang Zhang, Angjoo Kanazawa, QianqianWang. Proceedings of the Computer Vision and Pattern Recognition Conference, 2025

---

> ### Comment · Reviewer_fepN · 2025-11-27
>
> Thank the authors for the rebuttal. I will keep my positive scores.

---

### Author Response · Authors · 2025-12-03
**Review Summary**

We sincerely thank Area Chairs and the Reviewers for their time, and detailed, constructive, and insightful feedback. We especially appreciate the tremendous efforts of the newly assigned Area Chairs in carefully reviewing our rebuttal.

In the following overall summary, we provide a concise overview of the paper’s core contributions, summarize the reviewers’ feedback and main concerns, and highlight how each point was addressed and incorporated into the revised manuscript.

**Initial Reviews and Rebuttal Exchanges:** Our work received initial ratings of 6 (`fepN`), 6 (`WdWW`), 4 (`dzZe`), 4 (`c9w8`). Reviewer `dzZe` agreed to raise our score if all concerns are fully addressed but did not have time to respond to our rebuttal before the reviewer-response window closed.

**Key Strength:** The reviewers aligned on the key strengths of our work.

- **Originality in Multi-Control Video Synthesis.**
Reviewer `fepN` described our approach as "coherent and computationally attractive relative to MCMC‑based PoE sampling in prior work". `wdWW` highlighted our introduction of "a novel variational-inference framework for controllable video generation, which integrates multiple pretrained models into a unified and coherent compositional generative process". `c9w8` praised our system as a "conceptually interesting bridge" between heavy engineering composition and a sound theoretical framing.

- **Extensive and Rigorous Experiments.**
Reviewers `fepN`, `WdWW`, and `c9w8` praised our breadth and depth of our evaluation, including diverse settings, detailed ablations, comprehensive metrics (LPIPS, WorldScore, ViCLIP, GPT-4o), and strong qualitative results.

- **Clarity and Reproducibility.**
`WdWW` described the exposition as "clear and well-structured", and `dzZe` found the paper "well written and easy to follow", with `WdWW` noting that implementation details and reproducibility were well documented in the appendix.

- **Significance and Practical Impact.**
Reviewer `WdWW` emphasized that the work "addresses an important and timely challenge: combining multiple control modalities for coherent and physically consistent video generation", and `dzZe` noted it tackles "an important and practical problem of utilizing multiple expert models for controllable video synthesis".

- **Flexibility and Generality.**
`WdWW` noted that the proposed formulation is "general, training-free and can be applied on top of any modern diffusion or flow-based video model". `c9w8` highlighted the system’s "strong controllability and flexibility", supporting "multiple input modalities" in a unified generation process.

**Key Concerns and How we Addressed Them:**

> 1. Inference Cost and Runtime. (raised by `fepN`, `WdWW`, `dzZe`, and `c9w8`)

We expanded evaluation with detailed runtime and memory usage both in our rebuttal and revised manuscript Table S4. While our method is slow due to large pretrained models (Wan2.1 14B), 80% of the cost comes from denoising. However, our method is compatible with acceleration strategies such as timestep distillation, TeaCache, LoRA adapters, and multi-GPU parallelism.
> 2. Evaluation Scale and Generality. (raised by `fepN`)

We expanded the evaluation from 50 to 200 diverse test samples, covering 20 scenes, 8 camera trajectories, and 3 simulation categories (Table S1). In addition, a broader benchmark FullDit with 200 samples on camera to video (Table S2) supports stronger generalization claims.

> 3. Pipeline Complexity and Backbone Dependence (raised by `c9w8`)

We clarified that our modular, training-free design intentionally leverages multiple pretrained models, each contributing a distinct control capability. Components can be flexibly swapped as better backbones emerge. Experiments with various model combinations (e.g., Cam2V, Depth2V, Flow2V, Image2V) in original evaluation confirmed the robustness and adaptability of our approach.

> 4. Originality and Positioning (raised by `dzZe`, `c9w8`)

We clarified that our main contribution is a unified, inference-time framework for multi-controllable video generation, a capability not addressed by prior work. Rather than introducing a new architecture, we focus on compositional inference, enabling flexibility and generalization without retraining. Compared to unified training-based models, our approach (1) eliminates the need for costly multi-modal datasets and (2) naturally adapts to new user controls at inference time. To support this, we integrate variational inference with 3D-aware conditioning and demonstrate strong performance across 3D consistency, controllability, and visual quality.

We thank the Area Chairs and Reviewers again for their time and effort.

Best regards,

Authors of Submission #1114

---

### Meta-Review · Area_Chair_Bcwg · 2025-12-29

**Summary:**

This paper proposes a training-free framework that formulates controllable video synthesis as a variational inference problem, solving the challenge of integrating diverse, fine-grained user controls (text, trajectories, camera paths) into a unified generation process. By employing annealed SVGD and a 3D-aware context-conditioned factorization, the method effectively composes multiple pre-trained video backbones to ensure high controllability and 3D consistency without the need for retraining.
Reviewers commended the paper for its fine theoretical formulation and its extensive qualitative results, demonstrated superior controllability, 3D consistency, and visual fidelity compared to existing baselines.

Reviewer Concerns:
- **Inference Efficiency**: All reviewers raised significant concerns regarding the extremely high computational cost (~40 mins per video) and memory usage, questioning the practicality of the method for real-world applications (fepN, WdWW, dzZe, c9w8).
- **Novelty & Contribution**: Some reviewers perceived the method as an incremental engineering combination of existing pre-trained models rather than an architectural innovation, noting it is bounded by the capabilities of the chosen backbones (dzZe, c9w8).
- **Evaluation Scale & Scope**: Initial concerns were raised regarding the small size of the test set (50 samples) and the lack of sufficient comparison or analysis for long-video generation scenarios (fepN, dzZe).

In summary, this paper was reviewed by four experts in the field. The recommendations are 6, 6, 4, 4. The reviewers like the rigorous mathematical formulation and the impressive visual quality and flexibility of the proposed training-free framework. The reviewers raised the following concerns: the significant inference latency and the scale of the evaluation. After rebuttal, some concerns regarding the evaluation scale were solved by expanded experiments, while the efficiency and bounded capability remains a limitation.

**Reviewer Concerns:**

**Well addressed:**
- Evaluation Scale & Generality: The authors significantly expanded the test set from 50 to 200 samples, covering a wider range of scenes, camera trajectories, and simulation types. (fepN, dzZe)
- Justification of Novelty & Contribution: The authors clarified that their contribution lies in the ``a unified, inference-time framework`` rather than architectural design. They argued for the value of a training-free approach in avoiding the prohibitive costs of curating multi-modal paired datasets, which shifted reviewers' perception from incremental engineering to a valid design choice. (dzZe, c9w8)
- 3D Consistency & Inversion Robustness: The authors provided additional quantitative comparisons (MSE vs. DDIM inversion) and detailed explanations regarding 3D-aware context conditioning, satisfying questions about the method's sensitivity to inversion errors. (c9w8)

**Partly addressed:**
- Comparison with Long-Video Baselines: While the authors provided qualitative examples of long video generation and explained why direct quantitative comparison is difficult due to the lack of baselines supporting the same multi-control mixture, they did not provide the specific quantitative benchmarks against state-of-the-art long-video methods requested by the reviewer. (dzZe)

**Unsolved:**
- High Inference Cost & Efficiency: Although the authors provided the requested runtime statistics and suggested future acceleration paths, the fundamental issue remains: the method requires ~40 minutes per video segment on an H200 GPU. Reviewers maintained that this is a significant practical limitation that restricts the method to research demonstrations rather than deployment. (fepN, WdWW, dzZe, c9w8)
- Dependency on Backbone Capabilities: The concern that the method is conceptually bounded by the representational capacity and biases of the underlying pre-trained models remains an inherent characteristic of the framework that cannot be solved by rebuttal. (c9w8)

**Reviewer Scores:**

**fepN (6)**

The reviewer’s primary concerns regarding the small scale of the evaluation (originally 50 samples) and the lack of GPU usage statistics were fully addressed. The reviewer was satisfied with the clarifications on the equations and mask adaptation. The score would remain 6.

**WdWW (6)**

The concerns regarding the lack of runtime/memory statistics were resolved by the authors' new data tables.
The reviewer remained concerned about the high computational cost and scalability for industrial applications, but viewed the method's theoretical novelty and training-free nature as sufficient merits. The score likely holds 6.

**dzZe (4)**

The reviewer’s critique regarding incremental contribution and the limited experimental scope was addressed.
The request for quantitative comparisons against state-of-the-art long-video generation methods was only partly addressed.
dzZe explicitly stated that ``willing to increase my scores if my concerns are fully addressed.``, and the authors fulfilled the major request , it is likely dzZe would raise the score to 6.

**c9w8 (4)**

The reviewer explicitly maintained that the 40-minute inference time is a significant practical limitation, rendering the method impractical beyond research demos. c9w8 also noted the method remains bounded by the performance of the underlying backbones.
The score would remain 4.

---

### Decision · Program_Chairs · 2026-01-26

Reject